# Training-Free Text-Guided Color Editing with Multi-Modal Diffusion Transformer

**Zixin Yin[1,2]   Xili Dai[3]   Ling-Hao Chen[2,4]   Deyu Zhou[3,6]   Jianan Wang[5]   Duomin Wang[6]**
**Gang Yu[6]   Lionel M. Ni[1,3]   Lei Zhang[2]   Heung-Yeung Shum[1]**

[1] The Hong Kong University of Science and Technology
[2] International Digital Economy Academy
[3] The Hong Kong University of Science and Technology (Guangzhou)
[4] Tsinghua University   [5] Astribot   [6] StepFun

## Abstract

Text-guided color editing in images and videos is a fundamental yet unsolved problem, requiring fine-grained manipulation of color attributes, including albedo, light source color, and ambient lighting, while preserving physical consistency in geometry, material properties, and light-matter interactions. Existing training-free approaches provide broad applicability across editing tasks but struggle with precise color control and often introduce visual inconsistency in both edited and non-edited regions. In this work, we present ColorCtrl, a training-free color editing method that leverages the attention mechanisms of modern Multi-Modal Diffusion Transformers (MM-DiT). By disentangling structure and color through targeted manipulation of attention maps and value tokens, our method enables accurate and consistent color editing, along with word-level control of attribute intensity. Our method modifies only the intended regions specified by the prompt, leaving unrelated areas untouched. Extensive experiments on both SD3 and FLUX.1-dev demonstrate that ColorCtrl outperforms existing training-free approaches and achieves state-of-the-art performances in both edit quality and consistency. Furthermore, our method surpasses strong commercial models such as FLUX.1 Kontext Max and GPT-4o Image Generation in terms of consistency. When extended to video models like CogVideoX, our approach exhibits greater advantages, particularly in maintaining temporal coherence and editing stability. Finally, our method generalizes to instruction-based editing diffusion models such as Step1X-Edit and FLUX.1 Kontext dev, further demonstrating its versatility. Here is the website.

## 1 Introduction

Film industry-level color changing in images and videos based on textual instructions is a fundamental yet challenging task in deep learning and visual editing. Here, "color" encompasses not only object albedo (*i.e.*, intrinsic surface color independent of material properties), but also the color of light sources and ambient illumination. Color editing is an ill-conditioned problem, as it requires explicit or implicit 3D reconstruction of the entire scene, including correct illumination. During editing, it is essential to modify only the intended color attributes while preserving material properties, ensuring accurate reflections and refractions, and keeping non-editing regions unchanged.

Traditional image processing methods have been widely commercialized in professional software such as Photoshop, serving billions of users worldwide. However, the steep learning curve and significant manual effort required make it difficult to achieve widespread accessibility. Moreover, such tools are not well-suited for automated batch processing and cause problems when applied to video-related tasks. Recently, diffusion models have demonstrated remarkable capabilities in generating high-quality images that adhere to physical principles of color and illumination. This has spurred growing interest in leveraging their generative power to address the above challenges, with controllability emerging as a critical factor. Although many methods (Magar et al., 2025; Zhang et al., 2025) fine-tune diffusion models for controllable editing, they usually require large-scale datasets and complex training pipelines, and are often constrained to narrow domains or specific edit

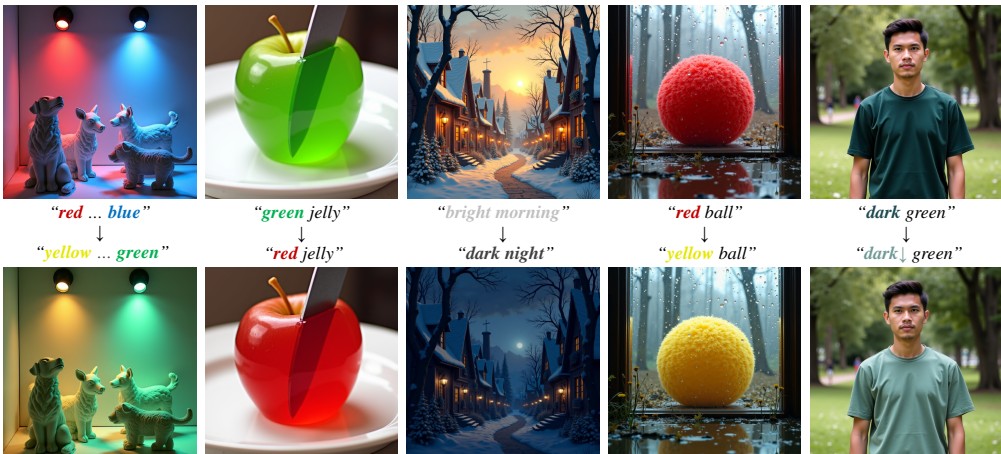

Figure 1: **Text-conditioned color editing.** Our method, ColorCtrl with FLUX.1-dev, edits colors across multiple materials while preserving light-matter interactions. For example, in the fourth case, the ball's color, its water reflection, specular highlights, and even small droplets on the glass have all been changed. It also enables fine-grained control over the intensity of specific descriptive terms.

types. On the other hand, training-free methods (Jiao et al., 2025) have gained popularity for a wide range of image editing tasks due to their generality and ease of use. Despite their success in many scenarios, they still struggle with fine-grained color control and often introduce inconsistencies in edited and non-edited regions, making precise color editing an unresolved challenge.

The recent architectural shift in diffusion models from U-Net (Rombach et al., 2022) to the Multi-Modal Diffusion Transformer (MM-DiT) (Esser et al., 2024) brings new opportunities for training-free color editing, for two main reasons: (1) the new architecture allows for scaling up both data and model capacity, enabling better adherence to physical priors; and (2) the improved fusion of text and vision modalities supports more flexible and precise attention-control editing strategies.

In this work, we propose **ColorCtrl**, a training-free, open-world color editing method that effectively leverages pre-trained MM-DiT models to perform natural and precise color modifications, while preserving all other visual attributes, such as geometry, material, reflection and refraction, light source position, illumination direction, and light intensity, as shown in Fig. 1. For example, when editing the color of a ball, ColorCtrl accurately adjusts not only the ball itself but also its reflection in the water, the specular reflections on both sides, and even the small water droplets on the glass surface, as demonstrated in the fourth example of Fig. 1. Furthermore, ColorCtrl supports fine-grained, word-level control over the strength of color attributes while preserving the other visual attributes mentioned above. Thanks to our precise and robust control, ColorCtrl requires no manual parameter tuning and can be directly applied across **all attention layers and inference timesteps**.

To verify the effectiveness of our method, we conduct extensive experiments showing that ColorCtrl outperforms existing training-free approaches on both Stable Diffusion 3 Medium (*a.k.a.,* SD3) (Esser et al., 2024) and FLUX.1-dev (Labs, 2024). To better understand the gap between open-source and commercial models, we further compare ColorCtrl with two strong commercial baselines: FLUX.1 Kontext Max (Black Forest Labs, 2025) and GPT-4o Image Generation (*a.k.a.*, GPT-4o) (OpenAI, 2025). ColorCtrl achieves more natural color editing and substantially better preservation of visual consistency, including background and structural fidelity. In addition, our method is model-agnostic and can be seamlessly extended to video models such as CogVideoX (Yang et al., 2024), where the performance gap becomes even more pronounced. Finally, ColorCtrl can be integrated into instruction-based editing models such as Step1X-Edit (Liu et al., 2025b) and FLUX.1 Kontext dev (Black Forest Labs, 2025), demonstrating strong compatibility.

In summary, the main contributions can be listed as follows.

- We propose **ColorCtrl**, a training-free method for color editing that supports modification of albedo, light source color and ambient lighting, while preserving the consistency of geometry, material and light-matter interaction.
- We present extensive experiments demonstrating that ColorCtrl achieves state-of-the-art performance among training-free methods on MM-DiT-based models. Compared to com-

mercial models (*i.e.*, FLUX.1 Kontext Max and GPT-4o Image Generation), our method delivers significantly better consistency preservation and produces more natural edits.

- ColorCtrl generalizes well across multiple MM-DiT-based models, including video and instruction-based editing models, highlighting its broad applicability and extensibility.

## 2 RELATED WORK

**Text-to-image and video generation.** Diffusion models with a U-Net backbone (Ho et al., 2020; Rombach et al., 2022; Guo et al., 2024) have largely replaced early GAN systems (Reed et al., 2016; Yu et al., 2023; Wang et al., 2023) due to superior image fidelity. However, the U-Net design scales poorly, leading to the adoption of Diffusion Transformers (DiT) (Peebles & Xie, 2023). Among these, MM-DiT (Esser et al., 2024) has emerged as a widely adopted backbone for recent state-of-the-art models (Esser et al., 2024; AI, 2024; Labs, 2024; Yang et al., 2024; Kong et al., 2024; Liu et al., 2025a; Ma et al., 2025; Black Forest Labs, 2025), such as SD3 (Esser et al., 2024) and FLUX.1-dev (Labs, 2024) for image generation, as well as CogVideoX (Yang et al., 2024) for video generation. In this work, we propose an attention control that plugs into any MM-DiT model.

**Text-guided editing.** Training-free text-guided editing methods based on pre-trained diffusion models offer flexibility and efficiency. Existing approaches fall into two groups: (i) *sampling-based* methods, which steer generation by injecting controlled noise or inversion (Jiao et al., 2025; Huberman-Spiegelglas et al., 2024; Xu et al., 2023; Kulikov et al., 2024; Yan et al., 2025); (ii) *attention-based* methods, which modify attention maps, starting with Prompt-to-Prompt (Hertz et al., 2023) and its image/video variants (Chen et al., 2024; Wang et al., 2025; Liu et al., 2024; Cao et al., 2023; Cai et al., 2025; Rout et al., 2025; Xu et al., 2025; Yin et al., 2025). Among these, DiTCtrl is the only method exploring attention control in MM-DiT, but ColorCtrl differs in key ways: DiTCtrl only applies mask extraction during long video generation, not editing, while our improved method is more robust. Additionally, DiTCtrl's re-weighting disrupts attention consistency, causing inconsistent geometries, and TextCrafter (Du et al., 2025) exhibits a similar issue, while ours avoids this. Furthermore, ColorCtrl works across all layers and timesteps, unlike DiTCtrl, which requires careful layer selection. Different from DiTCtrl, we operate in attention maps rather than key and value tokens. More recently, methods (Liu et al., 2025b; Brooks et al., 2023) like FLUX.1 Kontext (Black Forest Labs, 2025) and GPT-4o (OpenAI, 2025) have shown convenient creative workflows in training with synthetic instruction-response pairs for fine-tuning a diffusion model for image editing.

Despite advancements in text-guided editing, we demonstrate that current methods still face significant challenges in accurately changing colors while preserving geometry, material properties and light-matter interaction. Also, all aforementioned training-free methods rely on selectively manipulating specific inference steps or attention layers, which limits their robustness and consistency with respect to the source. In contrast, our approach requires no manual selection of steps or layers.

## 3 METHOD

We first formulate the color editing problem in Sec. 3.1, followed by a revisit of the attention mechanism in MM-DiT blocks in Sec. 3.2. Sec. 3.3 describes our approach for preserving geometry, material properties, and light-matter interactions. In Sec. 3.4, we introduce a method for preserving colors in non-editing regions, which are automatically detected by the model. Together, Sec. 3.3 and Sec. 3.4 constitute the core of our method. Finally, Sec. 3.5 introduces word-level control over color attributes via attention re-weighting, serving as an optional add-on for fine-grained attribute control.

### 3.1 TASK FORMULATION: TEXT-GUIDED COLOR EDITING

**Rendering process.** To provide a clearer and more precise description of the conditions required for film industry-level color changing, we define the rendering process of a frame by

$$\mathbf{I} = \mathcal{R}(G, L, A, S, C), \tag{1}$$

where the notations of rendering inputs are defined as,

- $\mathcal{R}(\cdot)$ - the rendering function,

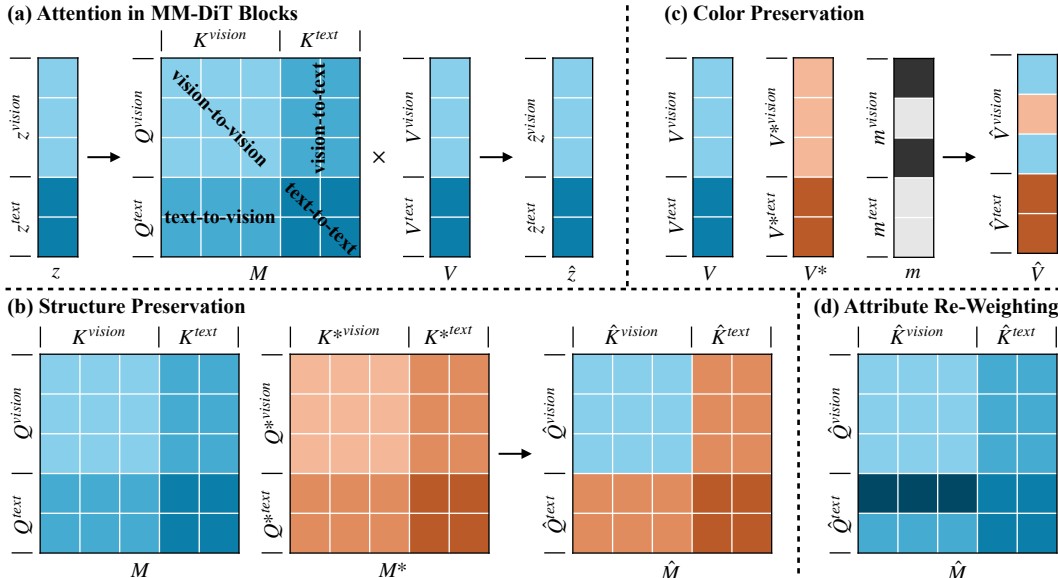

Figure 2: **Pipeline of ColorCtrl.** (a) Visualizes the attention mechanism in MM-DiT blocks. (b) Enables color editing while maintaining structural consistency. (c) Preserves colors in non-editing regions. (d) Applies attribute re-weighting to specific tokens. Symbols in the source branch have no superscript. Symbols with a superscript $^*$ indicate the target, and hats (*e.g.*, $\hat{V}$, $\hat{M}$) denote outputs.

- $G = \{G_1, \ldots, G_N\}$ - a set of object geometries, *e.g.*, mesh topology for $N$ objects,
- $L = \{L_{\text{env}}, L_1, \ldots, L_K\}$ - the ambient illumination $L_{\text{env}}$ and $K$ light sources (each with position, intensity, and spectrum),
- $A = \{A_1, \ldots, A_N\}$ - the albedos (base colors) of each object,
- $S = \{S_1, \ldots, S_N\}$ - material parameters (roughness, specularity, and normal maps) for each object that are color-independent,
- $C$ - camera intrinsics and extrinsics.

**Editing task.**  According to the predefined image rendering process in Eq. (1), we formulate the consistent editing task in this work as follows. Given a source image $\mathbf{I}$ and text prompt pairs $q$ before and after editing[1] that specifies which objects or lights to recolor and the desired target color, we aim to learn the following editing function $f(\cdot)$

$$f : (\mathbf{I}, q) \longmapsto \hat{\mathbf{I}} = \mathcal{R}(G, \hat{L}, \hat{A}, S, C), \tag{2}$$

such that $\hat{\mathbf{I}}$ satisfies:

(C1) **Geometry/view consistency** - $G$ and $C$ remain fixed with the source image $\mathbf{I}$, preserving object layout and perspective.

(C2) **Illumination consistency** - light positions and scalar intensities remain fixed. Specifically, changes may occur only on the target spectral components (*i.e.*, RGB channels) of $L$.

(C3) **Material consistency** - the object material $S$ remain fixed. The edits apply only to the albedo components $A$ of specific objects.

In the scope of the defined editing task in Eq. (2), the editing function $f(\cdot)$ needs to internally infer the underlying scene parameters and localize the editing objects, lights, and regions to perform precise, constrained color changes.

## 3.2 PRELIMINARIES: ATTENTION IN MM-DiT BLOCKS

The attention mechanism for text-vision fusion in MM-DiT (Esser et al., 2024) differs from that in U-Net (Rombach et al., 2022). In U-Net, cross-attention is used for text guidance, while self-attention

---

[1] An example of $q$: "white fox." $\rightarrow$ "orange fox." (source and target prompt respectively).

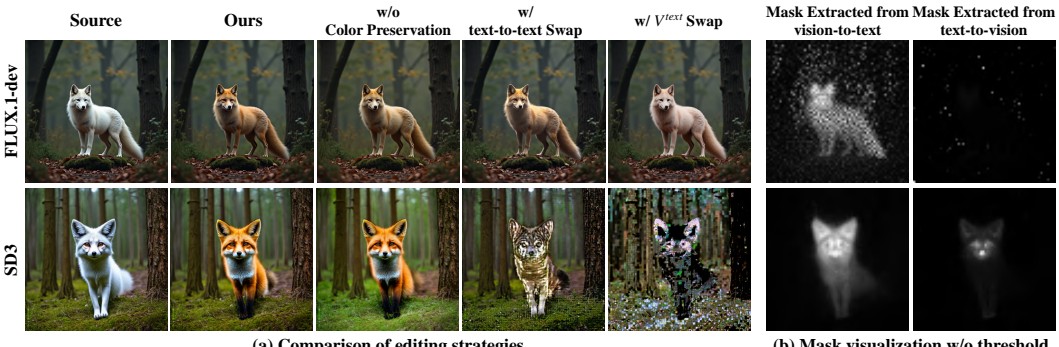

Figure 3: Top row: SD3 results; bottom row: FLUX.1-dev results. **(a)** The edit prompt is "white fox" → "orange fox". Left to right: source image, our full method, without color preservation, with swapped text-to-text part in structure preservation, and with swapped $V^{\text{text}}$ in color preservation. **(b)** The generation prompt is "a white fox in a forest", and the token for mask extraction is "fox". From left to right: the mask extracted from vision-to-text parts, and from text-to-vision parts.

focuses on visual content interactions, separately. In contrast, MM-DiT integrates text and vision by concatenating their tokens together and processing them jointly via sole self-attention. As illustrated in Fig. 2 (a), vision and text tokens $z$ are fed into the $i$-th MM-DiT block at timestep $t$ during inference. After modulation, the block produces an attention map $M$ and value tokens $V$, which are used to generate the updated tokens $\hat{z}$ for the next block and timestep. The resulting attention map $M$ can be divided into four parts: vision-to-vision (upper-left), vision-to-text (upper-right), text-to-vision (lower-left), and text-to-text (lower-right). These quadrants correspond to different query-key token pairings. For example, the text-to-vision region represents attention scores computed between query tokens from the text modality $Q^{\text{text}}$ and key tokens from the vision modality $K^{\text{vision}}$. Similarly, the value tokens $V$ are composed of a vision part $V^{\text{vision}}$ and a text part $V^{\text{text}}$. The functional roles of each region are discussed in detail in the following sections.

### 3.3 STRUCTURE PRESERVATION

Following Hertz et al. (2023); Cao et al. (2023), we divide the editing process into a source branch and a target branch. The source branch follows the original generation process, producing a source image and storing intermediate attention outputs. The target branch reuses these stored variables to generate edited results. Starting from a fixed random seed, the model can generate the desired object without applying any editing method. However, the resulting image often diverges significantly from the source, making it unsuitable for meaningful editing tasks that require structural consistency. To address this, all subsequent modifications are applied exclusively on the target branch, aiming to improve consistency without disrupting the intended edit. The process of keeping $G$, $S$, $C$, light positions, and scalar intensities fixed (as in constraints (C1)-(C3)) is referred to as **structure preservation**. We observe that the vision-to-vision part of the attention map inherently encodes rich knowledge about the parts of the scene that must remain unchanged. Given a source attention map $M$, its vision-to-vision part is transferred from $M$ to the corresponding part in the target attention map $M^*$, producing an edited attention map $\hat{M}$ that fully respects the structure preservation constraints, as described in Fig. 2 (b).

### 3.4 COLOR PRESERVATION

Despite enforcing structure preservation, we observe that undesired changes such as color shifts can still occur in regions unrelated to the edit, as shown in Fig. 3 (a). To further localize the edit and reduce inconsistencies, edits should be confined to the intended regions, while preserving all other areas. Motivated by the approach of Cai et al. (2025), we first extract a binary mask $m$ from vision-to-text parts of attention maps with a threshold $\epsilon$, which indicates the target editing region. In contrast to Cai et al. (2025), which averages the vision-to-text and text-to-vision parts to obtain the mask, we use only the vision-to-text parts, as they provide superior spatial localization for the target, unlike the text-to-vision parts (see Fig. 3 (b)). Based on $m$, we copy the value tokens from the non-editing regions in the vision part of the source $V$ into the corresponding regions of the target

Table 1: **Quantitative image results compared with training-free methods on PIE-Bench.** Results for FireFlow on SD3 are omitted due to their consistency being worse than using fixed seeds.

| Method | SD3 | | | | | FLUX.1-dev | | | | |
| --- | --- | --- | --- | --- | --- | --- | --- | --- | --- | --- |
| | Canny | BG Preservation | | Clip Similarity ↑ | | Canny | BG Preservation | | Clip Similarity ↑ | |
| | SSIM ↑ | PSNR ↑ | SSIM ↑ | Whole | Edited | SSIM ↑ | PSNR ↑ | SSIM ↑ | Whole | Edited |
| Fix seed | 0.5787 | 20.44 | 0.8411 | 29.17 | 27.54 | 0.7180 | 22.32 | 0.8877 | 27.72 | 25.76 |
| FireFlow (Deng et al., 2025b) | 0.6078 | 19.19 | 0.8461 | 28.63 | 27.24 | 0.8322 | 35.87 | 0.9717 | 25.84 | 23.56 |
| RF-Solver (Wang et al., 2025) | 0.6711 | 23.30 | 0.8906 | 27.43 | 25.96 | 0.8394 | 36.25 | 0.9715 | 25.83 | 23.58 |
| SDEdit (Meng et al., 2022) | 0.6353 | 27.78 | 0.8699 | 25.98 | 24.47 | 0.8285 | 32.62 | 0.9605 | 25.54 | 23.17 |
| DiTCtrl (Cai et al., 2025) | 0.8119 | 35.40 | 0.9812 | 26.21 | 24.67 | 0.8306 | 34.48 | 0.9791 | 25.89 | 23.58 |
| FlowEdit (Kulikov et al., 2024) | 0.7852 | 34.24 | 0.9704 | 26.13 | 24.97 | 0.8639 | 32.64 | 0.9774 | 25.95 | 23.63 |
| UniEdit-Flow (Jiao et al., 2025) | 0.8016 | 36.31 | 0.9774 | 26.08 | 24.67 | 0.8498 | 37.57 | 0.9777 | 25.78 | 23.44 |
| Ours | **0.8473** | **42.93** | **0.9960** | **28.32** | **26.96** | **0.9196** | **39.49** | **0.9936** | **27.34** | **24.90** |

$V^*$, yielding the final value tokens $\hat{V}$, as shwon in Fig. 2 (c). We refer to this procedure as **color preservation**. As shown in Fig. 3 (a), including the text part of value tokens $V^{\text{text}}$ during value exchange significantly weakens text guidance in FLUX.1-dev, and leads to severe artifacts in SD3.

### 3.5 ATTRIBUTE RE-WEIGHTING

So far, our method enables powerful color editing capabilities. However, the granularity of control via text prompts remains limited. For example, if the user specifies a color like "dark yellow", there is no way to explicitly control the degree of darkness. Existing re-weighting techniques in U-Net-based models primarily follow two approaches: (1) Scaling the text embedding of specific tokens before the diffusion process (Perry, 2023). However, this method is designed for CLIP-based (Radford et al., 2021) text encoders and is incompatible with MM-DiT-based models, which typically use T5 (Raffel et al., 2020). (2) Scaling the attention scores of specific tokens in the cross-attention map (Hertz et al., 2023). This is also inapplicable to MM-DiT, which relies solely on self-attention. Moreover, both existing approaches fail to maintain structural consistency during scaling, limiting their utility in high-fidelity editing tasks.

To support more fine-grained and user-friendly control in MM-DiT-based models, we introduce a mechanism to modulate the strength of specific attribute words. Specifically, we scale the attention scores in the text-to-vision parts of the attention map corresponding to the selected word tokens before the softmax operation, as illustrated in Fig. 2 (d). This modification can be applied either directly on the original source attention maps or on the target attention maps that have already undergone structure preservation, allowing flexible integration in above editing pipeline.

## 4 EXPERIMENTS

### 4.1 SETUP

**Baselines.** We compare our method against several training-free approaches built upon MM-DiT, including FireFlow (Deng et al., 2025b), RF-Solver (Wang et al., 2025), SDEdit (Meng et al., 2022), DiTCtrl (Cai et al., 2025), FlowEdit (Kulikov et al., 2024), and UniEdit-Flow (Jiao et al., 2025). We focus exclusively on MM-DiT-based baselines, as prior work (Jiao et al., 2025; Deng et al., 2025b) has shown that U-Net-based methods perform significantly worse. Accordingly, we exclude methods that cannot be adapted to the MM-DiT architecture. For commercial baselines, we compare with FLUX.1 Kontext Max (Black Forest Labs, 2025) and GPT-4o Image Generation (OpenAI, 2025).

**Implementation.** We conduct experiments on SD3 (Esser et al., 2024) and FLUX.1-dev (Labs, 2024) for image generation, and on CogVideoX-2B (Yang et al., 2024) for video generation. For instruction-based image editing, we use Step1X-Edit (Liu et al., 2025b) and FLUX.1 Kontext dev (Black Forest Labs, 2025). For FLUX.1-dev, Step1X-Edit, and FLUX.1 Kontext dev, we apply attention control to the single-stream attention layers, following Deng et al. (2025b). Unless otherwise noted, we use the Euler sampler and adopt UniEdit-Flow (Jiao et al., 2025) for image inversion. Maintaining a balance between fidelity to the original image and the strength of the applied edits is a well-known trade-off in generative editing. To ensure a fair comparison across methods, we carefully tune the hyperparameters for each baseline. Additional details are provided in Appendix A.

Table 2: Quantitative image results compared with commercial models on PIE-Bench.

| Method | SD3 | | | | | FLUX.1-dev | | | | |
|---|---|---|---|---|---|---|---|---|---|---|
| | Canny | BG Preservation | | Clip Similarity ↑ | | Canny | BG Preservation | | Clip Similarity ↑ | |
| | SSIM ↑ | PSNR ↑ | SSIM ↑ | Whole | Edited | SSIM ↑ | PSNR ↑ | SSIM ↑ | Whole | Edited |
| FLUX.1 Kontext Max (Black Forest Labs, 2025) | 0.7305 | 31.97 | 0.9254 | 28.93 | 27.30 | 0.7607 | 26.77 | 0.9165 | 28.36 | 26.10 |
| GPT-4o Image Generation (OpenAI, 2025) | 0.6240 | 23.69 | 0.8134 | **29.51** | **28.08** | 0.7431 | 23.71 | 0.8755 | **28.84** | **26.46** |
| Ours | **0.8473** | **42.93** | **0.9960** | 28.32 | 26.96 | **0.9196** | **39.49** | **0.9936** | 27.34 | 24.90 |

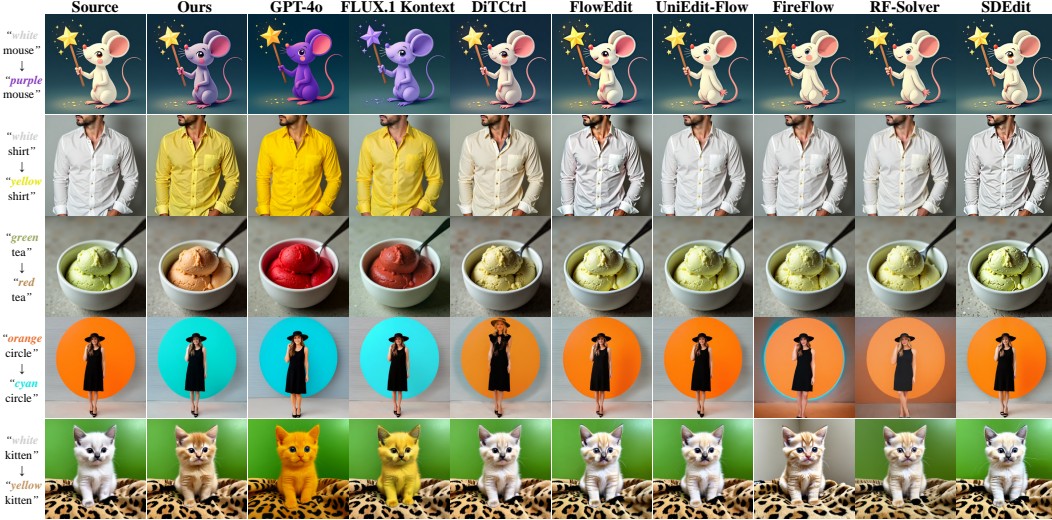

Figure 4: **Qualitative image results compared with training-free methods and commercial models on PIE-Bench.** The top three rows are generated using FLUX.1-dev, while the bottom two are generated using SD3. ***Best viewed with zoom-in.***

**Benchmark.** While prior editing methods (Hertz et al., 2023; Cao et al., 2023; Cai et al., 2025) typically lack standardized benchmark evaluation, we adopt prompts from the *Change Color* task in PIE-Bench (Ju et al., 2024), which includes 40 editing pairs, to better showcase the capabilities of our method. Although our approach is fully compatible with inversion methods, we adopt a noise-to-image setting during benchmark evaluations to better isolate and evaluate editing performance, removing the influence of reconstruction and inversion. This also allows us to reuse the same set of benchmark prompts for video diffusion models, enabling a consistent and fair comparison across both image and video domains. To further scale up the evaluation, we introduce a new benchmark called ColorCtrl-Bench, consisting of 300 prompt pairs in the same style as PIE-Bench (see Appendix A.4 for details). For all baselines, we adopt a fixed sampler and identical random seeds to ensure that source images are consistent, enabling reliable comparison across methods.

**Evaluation protocol.** Unlike the original PIE-Bench, which evaluates structural similarity using structural distance (Tumanyan et al., 2022), we adopt the Structural Similarity Index (SSIM) (Wang et al., 2004) computed on Canny edge maps (Canny, 1986), following the approach of Zhao et al. (2023), for more accurate assessment. To evaluate the preservation of non-edited regions (*a.k.a.*, BG Preservation), we compute PSNR and SSIM exclusively on those regions, which are annotated using Grounded SAM 2 (Ren et al., 2024) with dilation. Semantic alignment of the edits is assessed using CLIP similarity (Radford et al., 2021), applied to both the entire image and the edited regions.

### 4.2 COMPARISON WITH TRAINING-FREE METHODS (IMAGES)

Tab. 1 and Tab. 5 report benchmark results on both SD3 and FLUX.1-dev, comparing our method with other training-free baselines. Our method achieves **state-of-the-art** performance, delivering superior results in both preserving source content and executing accurate edits. Fig. 4 further supports this finding: other methods exhibit limited capacity for color editing and often introduce visual inconsistencies, while ours produces coherent and faithful edits. Additional results are in Appendix B.

### 4.3 Comparison with Commercial Models (Images)

Tab. 2 and Tab. 6 compare our method (based on SD3 and FLUX.1-dev) with two commercial models: FLUX.1 Kontext Max (Black Forest Labs, 2025) and GPT-4o Image Generation (OpenAI, 2025). Despite being based on open-source models, our approach achieves superior layout and detail consistency, as well as better preservation of non-edited regions. While the CLIP similarity scores of our method are slightly lower, visual results in Fig. 4 reveal that the commercial models often rely on **over-saturated, unrealistic edits** to better align with prompts. For example, in the top row, FLUX.1 Kontext Max recolors the entire mouse, including its magic wand, in solid purple, while GPT-4o produces a dark, dissonant shade. In contrast, our method applies a harmonious color. In the second row, only our method preserves the shirt's semi-transparency, whereas the commercial models render it as opaque yellow, ignoring material properties. In the third row, our method respects the muted tone of "green tea" and edits the ice cream to a natural reddish-brown "red tea" color. The commercial models, however, apply an unnaturally pure red. In the final row, although the prompt requests a "yellow kitten", no naturally occurring cat has a truly pure yellow coat. Our method instead generates a kitten with the closest plausible fur color, aligned with real-world appearances, unlike the commercial models, which apply flat, high-saturation tones that appear unnatural. These results demonstrate that higher CLIP similarity does not necessarily indicate better edit quality. It often stems from prompt overfitting while compromising realism and consistency. Overall, our method consistently produces more faithful, controllable edits, even built on open-source models.

### 4.4 Comparison with Training-Free Methods (Videos)

Thanks to the training-free nature of our method, it can be seamlessly extended to video editing tasks. Since FLUX.1 Kontext Max and GPT-4o do not support video editing, we compare only with other training-free methods. FlowEdit is excluded as it only applies to rectified flow models rather than diffusion-based models. Tab. 3 and Tab. 8 presents benchmark results on CogVideoX-2B (Yang et al., 2024).

Table 3: Quantitative video results compared with baselines on PIE-Bench.

| Method | Canny SSIM ↑ | BG Preservation PSNR ↑ | BG Preservation SSIM ↑ | Clip Similarity ↑ Whole | Clip Similarity ↑ Edited |
|---|---|---|---|---|---|
| Fix seed | 0.6228 | 20.73 | 0.8912 | 27.97 | 26.63 |
| FireFlow (Deng et al., 2025b) | 0.7517 | 30.74 | 0.9605 | 25.42 | 24.18 |
| RF-Solver (Wang et al., 2025) | 0.7677 | 35.41 | 0.9730 | 25.40 | 24.08 |
| SDEdit (Meng et al., 2022) | 0.7880 | 35.46 | 0.9689 | 24.64 | 23.21 |
| DiTCtrl (Cai et al., 2025) | 0.6912 | 27.70 | 0.9435 | 27.02 | 25.57 |
| UniEdit-Flow (Jiao et al., 2025) | 0.7645 | 38.00 | 0.9772 | 25.35 | 24.13 |
| Ours | **0.8651** | **38.98** | **0.9916** | **27.12** | **25.96** |

Similar to image editing, our method outperforms all baselines. Notably, the performance gap becomes even more pronounced due to the added temporal dimension. Visualization results in Fig. 6 further highlight the effectiveness of ColorCtrl. Consistent with its performance in image editing, our method also handles challenging cases in video, such as accurately reflecting the color change of the ice cream in the bowl's reflection. Please refer to Appendix B for additional comparisons.

### 4.5 Attribute Re-Weighting Analysis

Although DiTCtrl (Cai et al., 2025) includes a mechanism for attribute re-weighting, both the paper and results in Tab. 1 suggest limitations. Specifically, the method struggles to achieve the intended color changes while maintaining consistency with the source image. Moreover, their approach of scaling attention weights after the softmax operation violates the assumption that attention scores should sum to one, which leads to incorrect attention behavior. Since none of the baselines can smoothly adjust attribute strength while simultaneously satisfying the three constraints ((C1)-(C3)), we present results of our method alone in Fig. 5. These results show that our method not only supports re-weighting a single attribute within the same prompt, but also allows adjusting attribute

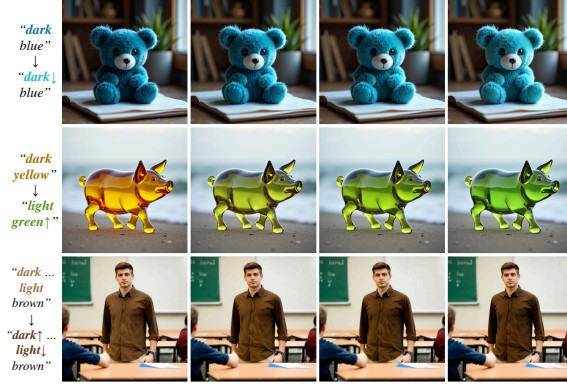

Figure 5: **Examples of attribute re-weighting.** The top two rows are generated using FLUX.1-dev, while the bottom one are generated using SD3.

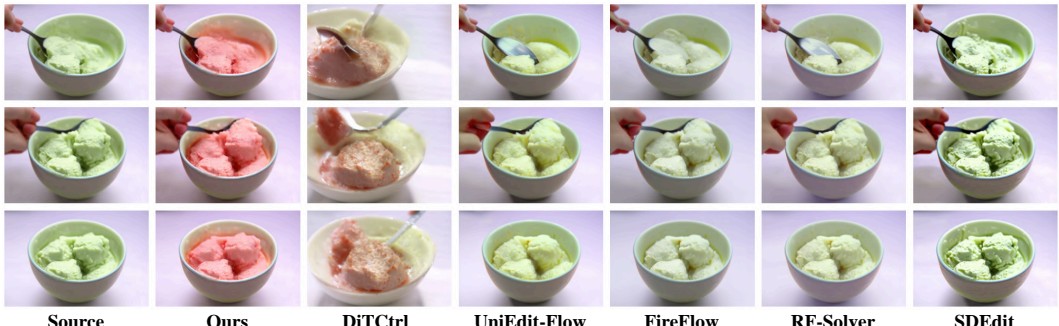

| Source | Ours | DiTCtrl | UniEdit-Flow | FireFlow | RF-Solver | SDEdit |

Figure 6: **Qualitative video results compared with training-free methods on PIE-Bench.** The edit prompt is "green tea" → "red tea". Each shows three frames.

strength across different prompts (second row). Moreover, our method can re-weight multiple attributes simultaneously (third row). Overall, these results demonstrate that ColorCtrl enables smooth and controllable transitions in attribute strength, while preserving structural consistency across the image and maintaining color fidelity in non-edited regions, on both SD3 and FLUX.1-dev.

### 4.6 ABLATION STUDY

According to Tab. 4, starting from fixed random seed generation, we observe the highest CLIP similarity due to the lack of consistency constraints. However, this comes at the cost of very low scores in Canny SSIM, as well as PSNR and SSIM in non-edited regions, indicating poor structural and visual consistency. Introducing the structure preservation component significantly improves all consistency metrics, suggesting that the geometric and material attributes are effectively maintained, as also illustrated in Fig. 3. Adding the color preservation component completes our method, which further enhances consistency, particularly in non-edited regions, while sacrificing almost no CLIP similarity. Overall, the results validate the effectiveness of each component in our method.

### 4.7 COMPATIBILITY AND DISCUSSION

**Real image editing.** To apply ColorCtrl to real input images, we integrate the inversion method from UniEdit-Flow (Jiao et al., 2025) and replace the original editing module with our method. As shown in Fig. 7, our approach generalizes well to real-world inputs and matches its noise-to-image performance on both SD3 and FLUX.1-dev, preserving fine-grained consistency (*e.g.*, subtle fabric wrinkles and shadows) while delivering strong editing performance. Notably, in the top row, even when editing black clothing, our method accurately distinguishes material shading from cast shadows, resulting in illumination-consistent edits.

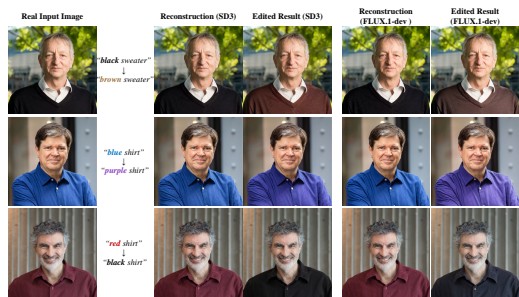

Figure 7: **Examples of real image editing.** Results generated with SD3 (left) and FLUX.1-dev (right).

**Generalization to instruction-based editing diffusion models.** In addition to text-to-image and text-to-video models, ColorCtrl is also compatible with instruction-based editing diffusion models, such as Step1X-Edit (Liu et al., 2025b) and FLUX.1 Kontext dev (Black Forest Labs, 2025). Given a real input image and a target editing instruction, the model performs edits accordingly. However, performing a second round of editing using the original model alone often leads to structural inconsistencies, such as distortions or shifts in shadows and edges. By incorporating our method, the model can further refine color edits while preserving structural fidelity in both edited and non-edited

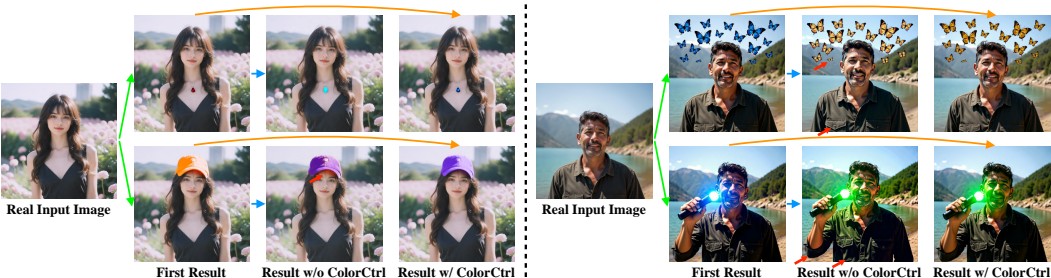

Figure 8: **Examples of results generated with Step1X-Edit (left) and FLUX.1 Kontext dev (right).** Green arrows: first edit using the editing model. Blue arrows: second edit directly using the editing model conditioned on the result of the first edit. Orange arrows: second edit using the editing model with ColorCtrl. Top left: a red diamond is added to the neck, then changed to blue. Bottom left: an orange cap is added, then changed to purple. Top right: blue butterflies are added, then changed to yellow. Bottom right: a flashlight with blue light is added, then changed to green.

Table 4: Ablation study evaluating the effectiveness of each component on PIE-Bench.

| Method | SD3 | | | | | FLUX.1-dev | | | | |
|---|---|---|---|---|---|---|---|---|---|---|
| | Canny | BG Preservation | | Clip Similarity ↑ | | Canny | BG Preservation | | Clip Similarity ↑ | |
| | SSIM ↑ | PSNR ↑ | SSIM ↑ | Whole | Edited | SSIM ↑ | PSNR ↑ | SSIM ↑ | Whole | Edited |
| Fix seed | 0.5787 | 20.44 | 0.8411 | **29.17** | **27.54** | 0.7180 | 22.32 | 0.8877 | **27.72** | **25.76** |
| + Structure Preservation | 0.7312 | 24.77 | 0.9201 | 28.41 | 27.29 | 0.9019 | 30.20 | 0.9719 | 27.44 | 25.22 |
| + Color Preservation (Ours) | **0.8473** | **42.93** | **0.9960** | 28.32 | 26.96 | **0.9196** | **39.49** | **0.9936** | 27.34 | 24.90 |

regions. As shown in Fig. 8, compared to using the base model alone, our approach achieves better outline consistency and improved preservation of subtle visual cues like shadows.

**Additional results.** Appendix B presents benchmark evaluations on ColorCtrl-Bench, limitation analysis, user studies, and downstream applications.

**Discussion.** Our method integrates structure preservation, regional color preservation, and word-level attribute intensity control into a unified, training-free pipeline. These components work together to enable precise and consistent text-driven color editing. Each part of the attention computation plays a distinct role: the vision-to-vision part of $M$ preserves structure, the vision-to-text part is used for mask extraction, the text-to-vision part enables attribute re-weighting, the vision part of $V$ supports color preservation, and the text-to-text region of $M$ along with the text part of $V$ provides crucial textual guidance. As demonstrated by the degradation observed in Fig. 3 (a) when the text-to-text region of $M$ or the text part of $V$ is altered, preserving the integrity of these parts is essential for maintaining robust and coherent generation that aligns with the target prompt.

## 5 CONCLUSION

We have introduced **ColorCtrl**, a training-free method for text-guided color editing that achieves fine-grained, physically consistent control over albedo, light source color, and ambient illumination. Our method is designed to edit only the intended visual attributes specified in the prompt, leaving all unrelated regions untouched. Built upon diverse MM-DiT-based diffusion models, such as SD3 and FLUX.1-dev, our approach enables fine-grained control over attribute intensity, while preserving geometry, material properties, and light-matter interactions. ColorCtrl not only outperforms prior training-free methods and achieves state-of-the-art results but also delivers stronger consistency than commercial models, *i.e.*, FLUX.1 Kontext Max and GPT-4o Image Generation, on both quantitative and qualitative evaluations. Moreover, its model-agnostic design generalizes naturally to video diffusion models (*i.e.*, CogVideoX) and instruction-based editing diffusion models (*i.e.*, Step1X-Edit and FLUX.1 Kontext dev), highlighting its broad applicability. We believe ColorCtrl paves the way for scalable, high-fidelity, and controllable color editing in both research and practical deployment.

ETHICS STATEMENT

Advances in image editing technologies bring not only new capabilities but also ethical challenges. While our method improves the precision of color editing via text-based control, it may also be misapplied to produce deceptive or harmful visual content. To mitigate such risks, we advocate for responsible usage practices that prioritize transparency, user awareness, and informed consent in real-world applications. Moreover, since our method relies on pretrained models, it inherits any biases embedded in these models. Such biases may surface in the edited outputs in subtle or unintended ways. We consider this an open area for further investigation and support ongoing efforts to identify and reduce algorithmic bias. All human studies were conducted with voluntary participants who were informed of the goals of the study and provided explicit consent.

REPRODUCIBILITY STATEMENT

To support reproducibility, we provide comprehensive details regarding the inference process and evaluation protocols in Sec. 3, Sec. 4.1, and Appendix A. We are committed to releasing the complete source code, including implementation scripts and necessary dependencies, upon acceptance. This will allow other researchers to reproduce our experiments and extend ColorCtrl in future work.

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

# A  IMPLEMENTATION DETAILS

## A.1  INFERENCE SETTINGS

For benchmarking, we use 28 inference steps for both SD3 (Esser et al., 2024) and FLUX.1-dev, with the classifier-free guidance (CFG) scale (Ho & Salimans, 2021) set to 7.5. All images are generated at a resolution of $1024 \times 1024$. For CogVideoX-2B, we use 50 inference steps and set the CFG scale to 6, generating videos at 49 frames with $720 \times 480$ resolution. A fixed random seed of 42 is used for all benchmark experiments. For real image editing, we use the latest inversion method from UniEdit-Flow (Jiao et al., 2025) and set the CFG scale to 1 according to the method.

The target object for mask extraction is determined using the "blended_word" keywords provided by PIE-Bench (Ju et al., 2024) and ColorCtrl-Bench. We use a mask threshold of $\epsilon = 0.1$ which is the same as DiTCtrl, which consistently yields strong performance across SD3, FLUX.1-dev, and CogVideoX-2B. Despite being relatively coarse, this threshold is sufficient due to the strong global adaptation ability of modern generative models, which enables them to propagate edits from sparse color cues to semantically aligned regions.

For Step1X-Edit (Liu et al., 2025b) and FLUX.1 Kontext dev (Black Forest Labs, 2025), we use their official code and setting with resolution $1024 \times 1024$.

Image generation is performed on an RTX 4090 GPU, while video generation uses an A100 GPU.

## A.2  SAMPLING DETAILS

To avoid redundant computation and accelerate inference, the source branch is first executed to cache attention maps and value tokens for reuse, following a similar strategy to that in Wang et al. (2025). During this stage, the editing mask $m$ is also computed to indicate the regions to be modified. In the actual editing phase, the cached features and mask are directly loaded, eliminating the need to recompute the source branch. This approach ensures that the editing process remains as efficient as standard sampling methods, without introducing any additional computational cost.

## A.3  IMPLEMENTATION OF COMPARED METHODS

Several compared methods lack official implementations for SD3 or CogVideoX-2B, or do not provide compatible sampling code. To ensure fair comparison, we reimplement these methods on SD3, FLUX.1-dev, and CogVideoX-2B by faithfully following their original designs and carefully tuning hyperparameters to match the reported performance. Detailed implementations are as follows:

- **DiTCtrl (Cai et al., 2025)**: For SD3-based image editing, we set the editing range from timestep 2 to 17, modifying the last 5 blocks. For FLUX.1-dev, we edit from timestep 2 to 11 across the last 6 blocks. For video editing with CogVideoX-2B, the official implementation is used. During editing, key ($K$) and value ($V$) tokens from the source branch are copied into the attention layers of the target branch.
- **FlowEdit (Kulikov et al., 2024)**: We adopt the official image editing setting for FLUX-1-dev. For SD3, we rescale the $n\_max$ factor to account for the change in inference steps. Specifically, since the number of steps has changed from 50 to 28, we set $n\_max = 28 \div 50 \times 33 \approx 18$, while keeping all other parameters unchanged.
- **UniEdit-Flow (Jiao et al., 2025)**: The official version supports SD3 and FLUX.1-dev but provides the $\omega$ parameter only for CFG = 1. Following the similarity transformation introduced in the paper, we use $\omega = 5 \div 7.5 \approx 0.6$ and set $\alpha = 0.6$ for SD3, $\alpha = 0.85$ for FLUX.1-dev, and $\alpha = 0.8$ for video generation, which yields comparable performance to the original.
- **FireFlow (Deng et al., 2025b)**: In SD3, we observe that it is difficult to select a suitable end timestep for FireFlow, as setting it too high often leads to artifacts and generation failures, as shown in Fig. 9. To mitigate quality degradation caused by excessive editing steps, we restrict editing to timesteps 0 through 3 across all blocks in SD3. In contrast, FLUX.1-dev does not exhibit this issue, and using the same range (timesteps 0 to 3) yields results comparable to those reported in the original paper. For video generation, the editing ends at timestep 9. During editing, value tokens ($V$) are copied from the source to the target.

- **RF-Solver (Wang et al., 2025)**: RF-Solver exhibits a similar issue to FireFlow when applied to SD3, where higher editing timesteps lead to artifacts and degraded generation quality, as illustrated in Fig. 9. To address this, we limit editing to timesteps 0 to 7 in the second half of the SD3 blocks. For FLUX.1-dev, we set the end timestep to 4, which yields stable results without noticeable artifacts. Value tokens ($V$) are copied from the source to the target. For video generation, editing is performed up to timestep 9.
- **SDEdit (Meng et al., 2022)**: We set $t_0 = 0.6$ and apply editing to both generated and real input images or videos. The same parameter is applied across SD3, FLUX.1-dev, and CogVideoX-2B.
- **FLUX.1 Kontext Max (Black Forest Labs, 2025)**: Results are obtained using the official API, with source images and instruction prompts taken from PIE-Bench (Ju et al., 2024).
- **GPT-4o Image Generation (OpenAI, 2025)**: Results are generated through the official client using the source images and instruction prompts from PIE-Bench.

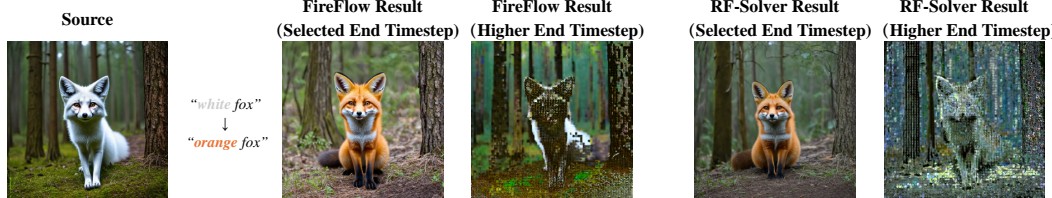

Figure 9: **Examples of FireFlow and RF-Solver with difference end timesteps on SD3.** The selected end timestep refers to the setting used in the benchmark evaluation, while the higher end timestep denotes a larger value chosen for comparison purposes.

### A.4   COLORCTRL-BENCH CONSTRUCTION

Here, we provide details on the construction of ColorCtrl-Bench. Following a similar approach to that in Ju et al. (2024), we use GPT to generate a dataset of tuples, each consisting of a source prompt, a target prompt, a subject token, and an instruction, consistent with the format used in PIE-Bench. These tuples represent an image before and after a color editing operation, along with an instruction describing the transformation and a subject token indicating the object to be modified. The exact prompting template is shown in Fig. 23.

### A.5   EVALUATION DETAILS

For editing region masks, we first detect the target object with Grounded SAM 2 (Ren et al., 2024), using the "blended_word" in the PIE-Bench and ColorCtrl-Bench as the detection keyword. The raw mask is then dilated to compensate for the uncertainty of the boundary, slightly enlarging the edited region. A marginally eroded mask is still adequate for BG preservation metrics, and a marginally dilated foreground does not bias CLIP similarity scores. Hence, this dilation neither undermines the consistency check nor suffers from the boundary instability of Grounded SAM 2. For video evaluation, we compute each metric frame-wise and report the average over all frames as the final score for the video.

## B   MORE RESULTS AND ANALYSIS

### B.1   DOWNSTREAM APPLICATION: SYNTHETIC COLOR EDITING AND FINE-TUNING

We generate 40k pairs of color-editing examples by applying ColorCtrl to FLUX.1-dev, and use this synthetic corpus to fine-tune Step1X-Edit-v1.1 (Liu et al., 2025b) for 2,300 H800 GPU-hours. This downstream model shows clear gains on color-editing tasks, especially under complex light–matter interactions such as reflections and refractions, while better preserving object details. Leveraging the multilingual capability of Step1X-Edit-v1.1, we also compare the original and fine-tuned models with Chinese edit instructions, as shown in Fig. 10. This highlights our potential of building pairwise color editing data in the future.

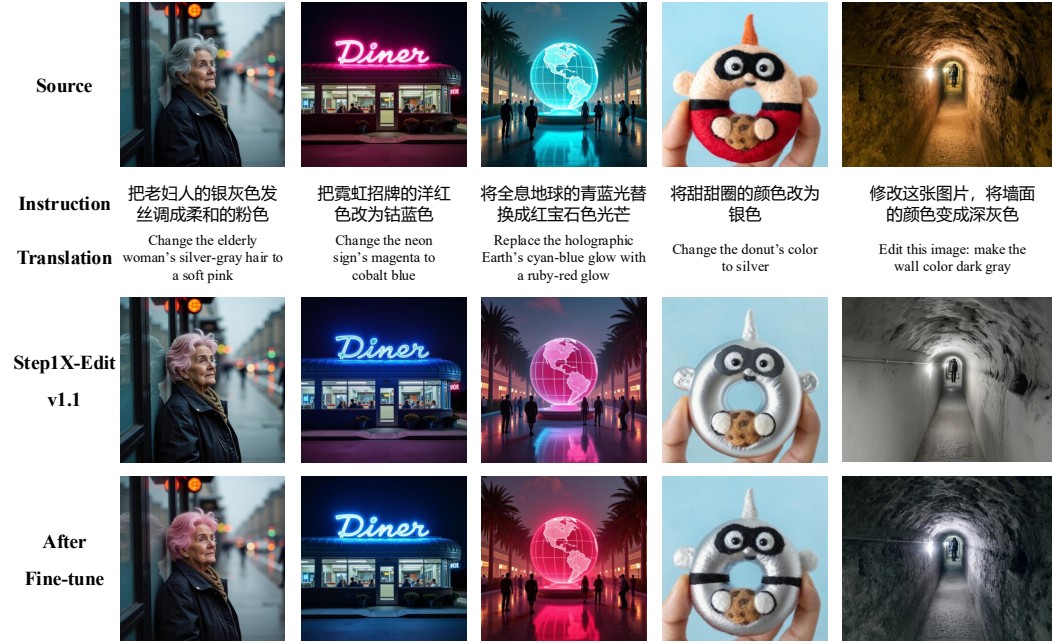

Figure 10: **Step1X-Edit-v1.1 fine-tuned on ColorCtrl-generated color-editing data vs. original.** Results are produced with Chinese edit instructions and English translations are shown below.

Table 5: **Quantitative image results compared with training-free methods on ColorCtrl-Bench.** Results for FireFlow on SD3 are omitted due to consistency worse than using fixed seeds.

| Method | SD3 | | | | | FLUX.1-dev | | | | |
| | Canny | BG Preservation | | Clip Similarity ↑ | | Canny | BG Preservation | | Clip Similarity ↑ | |
| | SSIM ↑ | PSNR ↑ | SSIM ↑ | Whole | Edited | SSIM ↑ | PSNR ↑ | SSIM ↑ | Whole | Edited |
|---|---|---|---|---|---|---|---|---|---|---|
| Fix seed | 0.6805 | 16.93 | 0.8303 | 28.27 | 26.65 | 0.7569 | 19.54 | 0.8552 | 27.70 | 26.25 |
| FireFlow (Deng et al., 2025b) | 0.6625 | 13.87 | 0.7664 | 27.45 | 25.95 | 0.8553 | 32.80 | 0.9594 | 25.05 | 23.69 |
| RF-Solver (Wang et al., 2025) | 0.7329 | 18.69 | 0.8596 | 26.70 | 25.19 | 0.8832 | 36.25 | 0.9736 | 24.89 | 23.49 |
| SDEdit (Meng et al., 2022) | 0.7643 | 27.24 | 0.9287 | 25.22 | 23.77 | 0.8430 | 29.47 | 0.9428 | 24.45 | 22.83 |
| DiTCtrl (Cai et al., 2025) | 0.8465 | 31.65 | 0.9723 | 25.25 | 23.82 | 0.8699 | 31.72 | 0.9707 | 24.90 | 23.53 |
| FlowEdit (Kulikov et al., 2024) | 0.8261 | 30.77 | 0.9622 | 25.53 | 24.24 | 0.8704 | 29.29 | 0.9620 | 25.09 | 23.52 |
| UniEdit-Flow (Jiao et al., 2025) | 0.8503 | 32.74 | 0.9725 | 25.24 | 23.81 | 0.8725 | 34.85 | 0.9676 | 25.04 | 23.61 |
| Ours | **0.8775** | **38.16** | **0.9896** | **28.07** | **26.69** | **0.9324** | **37.96** | **0.9901** | **26.53** | **25.26** |

## B.2 USER STUDY

A total of 24 expert participants conducted pairwise evaluations on 34 image cases from PIE-Bench and ColorCtrl-Bench. For each comparison, presentation order placement was randomized and method identities were blinded. Raters chose the preferred result according to a predefined rubric (edit success, naturalness, background preservation). Overall, ColorCtrl was preferred in 64.71% of comparisons, outperforming all baselines (Tab. 7). The user-study interface is shown in Fig. 22.

## B.3 MORE RESULTS OF IMAGE EDITING

Tab. 5 and Tab. 6 present quantitative results of image editing tasks on ColorCtrl-Bench, comparing our method against both training-free baselines and commercial models. Visual examples are shown in Fig. 17. Consistent with our findings on PIE-Bench, ColorCtrl outperforms other methods in maintaining object consistency and achieves accurate color edits with proper illumination and reflection. The similar conclusion drawn from this larger-scale benchmark further highlights the robustness and effectiveness of our approach.

Fig. 18 presents additional image editing results on PIE-Bench, comparing our method with both training-free and commercial models. In the first column, only ColorCtrl successfully preserves the texture of the rocks, while other methods either fail to change the color or introduce texture distor-

Table 6: Quantitative image results compared with commercial models on ColorCtrl-Bench.

| Method | SD3 | | | | | FLUX.1-dev | | | | |
|---|---|---|---|---|---|---|---|---|---|---|
| | Canny | BG Preservation | | Clip Similarity ↑ | | Canny | BG Preservation | | Clip Similarity ↑ | |
| | SSIM ↑ | PSNR ↑ | SSIM ↑ | Whole | Edited | SSIM ↑ | PSNR ↑ | SSIM ↑ | Whole | Edited |
| FLUX.1 Kontext Max (Black Forest Labs, 2025) | 0.8016 | 30.40 | 0.9152 | 28.23 | **26.83** | 0.8032 | 27.18 | 0.8854 | 27.56 | **26.24** |
| GPT-4o Image Generation (OpenAI, 2025) | 0.6988 | 21.69 | 0.8030 | **28.24** | 26.66 | 0.7727 | 23.05 | 0.8371 | **27.65** | **26.24** |
| Ours | **0.8775** | **38.16** | **0.9896** | 28.07 | 26.69 | **0.9324** | **37.96** | **0.9901** | 26.53 | 25.26 |

Table 7: User study results.

| Method | Preference (%) |
|---|---|
| SDEdit (Meng et al., 2022) | 0 |
| RF-Solver (Wang et al., 2025) | 0 |
| FireFlow (Deng et al., 2025b) | 0 |
| UniEdit-Flow (Jiao et al., 2025) | 0 |
| FlowEdit (Kulikov et al., 2024) | 0 |
| DiTCtrl (Cai et al., 2025) | 0 |
| FLUX.1 Kontext Max (Black Forest Labs, 2025) | 23.53 ± 14.00 |
| GPT-4o Image Generation (OpenAI, 2025) | 11.76 ± 8.31 |
| Ours | **64.71** ± 9.90 |

Table 8: Quantitative video results compared with baselines on ColorCtrl-Bench.

| Method | Canny | BG Preservation | | Clip Similarity ↑ | |
|---|---|---|---|---|---|
| | SSIM ↑ | PSNR ↑ | SSIM ↑ | Whole | Edited |
| Fix seed | 0.6671 | 19.49 | 0.8489 | 28.01 | 26.66 |
| FireFlow (Deng et al., 2025b) | 0.7957 | 28.78 | 0.9323 | 25.26 | 24.12 |
| RF-Solver (Wang et al., 2025) | 0.8102 | 32.64 | 0.9562 | 25.36 | 24.16 |
| SDEdit (Meng et al., 2022) | 0.8281 | 34.30 | 0.9626 | 24.74 | 23.46 |
| DiTCtrl (Cai et al., 2025) | 0.7270 | 26.14 | 0.9165 | 26.54 | 25.14 |
| UniEdit-Flow (Jiao et al., 2025) | 0.8002 | 35.33 | 0.9613 | 25.39 | 24.14 |
| Ours | **0.8885** | **38.68** | **0.9893** | **27.32** | **26.16** |

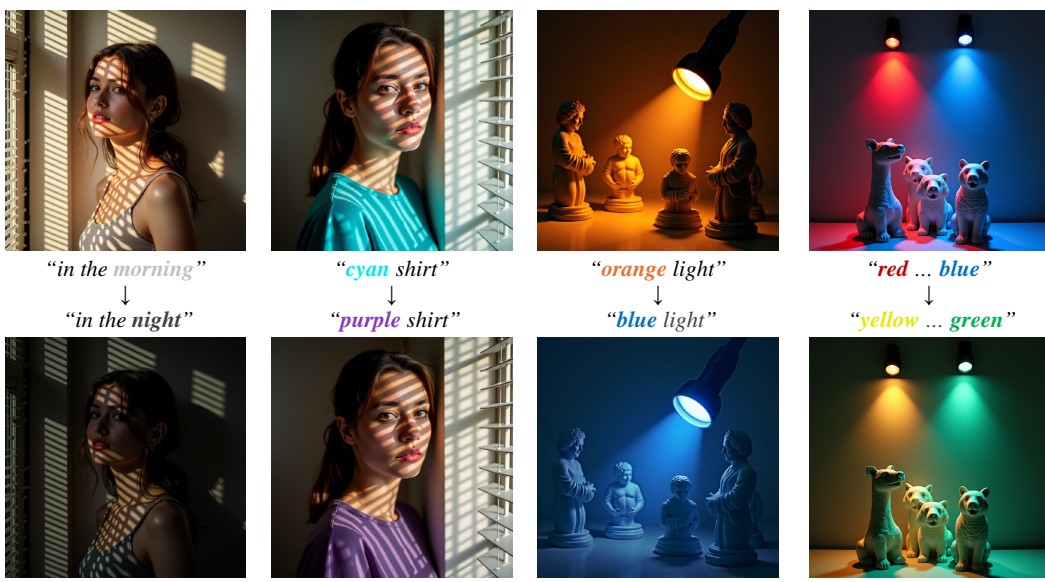

*"in the morning"* → *"in the night"*  *"cyan shirt"* → *"purple shirt"*  *"orange light"* → *"blue light"*  *"red … blue"* → *"yellow … green"*

Figure 11: Examples of image editing results.

tions. In the third column, FLUX.1 Kontext Max and GPT-4o alter the pose and fabric wrinkles, whereas other training-free methods fail to perform the intended edit.

Fig. 11 shows additional image results with FLUX.1-dev.

### B.4 MORE RESULTS OF VIDEO EDITING

Tab. 8 shows quantitative results of video editing tasks on ColorCtrl-Bench. Fig. 19 and Fig. 20 present video editing results on ColorCtrl-Bench, while Fig. 21 shows additional results on PIE-Bench. Our method consistently outperforms all baselines, achieving superior color editing accuracy and better structural consistency across frames.

### B.5 ADDITIONAL EVALUATION PROTOCOL

#### B.5.1 STRUCTURAL-PRESERVATION METRICS

Beyond the Canny-SSIM metric used in the main paper, we report two additional gradient-based metrics to further verify edge and fine-detail consistency under strong color edits:

Table 9: Additional quantitative image results compared with training-free methods on PIE-Bench.

| Method | SD3 | | | | FLUX.1-dev | | | |
|---|---|---|---|---|---|---|---|---|
| | Y-GCS ↑ | Y-GMS ↑ | SC ↑ | PQ ↑ | Y-GCS ↑ | Y-GMS ↑ | SC ↑ | PQ ↑ |
| Fix seed | 0.5430 | 0.6978 | 5.300 | 7.475 | 0.4905 | 0.7579 | 4.775 | 8.325 |
| FireFlow (Deng et al., 2025b) | 0.6105 | 0.7271 | 5.600 | 7.650 | 0.8020 | 0.9107 | 1.875 | 8.550 |
| RF-Solver (Wang et al., 2025) | 0.7339 | 0.7891 | 5.525 | 8.075 | 0.8068 | 0.9127 | 1.825 | 8.675 |
| SDEdit (Meng et al., 2022) | 0.7339 | 0.8109 | 1.950 | 7.975 | 0.8797 | 0.8967 | 0.350 | 7.900 |
| DiTCtrl (Cai et al., 2025) | 0.9074 | 0.9281 | 2.000 | 7.925 | 0.8553 | 0.9049 | 2.275 | 8.425 |
| FlowEdit (Kulikov et al., 2024) | 0.9027 | 0.9183 | 2.800 | 7.925 | 0.9061 | 0.9337 | 1.625 | 8.175 |
| UniEdit-Flow (Jiao et al., 2025) | 0.9034 | 0.9264 | 2.075 | 8.150 | 0.8320 | 0.9261 | 1.825 | 8.675 |
| Ours | **0.9244** | **0.9316** | **7.725** | **8.350** | **0.9555** | **0.9467** | **7.100** | **9.025** |

Table 10: Additional quantitative image results on PIE-Bench compared with commercial models on PIE-Bench.

| Method | SD3 | | | | FLUX.1-dev | | | |
|---|---|---|---|---|---|---|---|---|
| | Y-GCS ↑ | Y-GMS ↑ | SC ↑ | PQ ↑ | Y-GCS ↑ | Y-GMS ↑ | SC ↑ | PQ ↑ |
| Fix seed | 0.5430 | 0.6978 | 5.300 | 7.475 | 0.4905 | 0.7579 | 4.775 | 8.325 |
| FLUX.1 Kontext Max (Black Forest Labs, 2025) | 0.7533 | 0.8104 | 8.750 | 7.393 | 0.5674 | 0.8079 | 8.525 | 7.625 |
| GPT-4o Image Generation (OpenAI, 2025) | 0.5901 | 0.7435 | **8.923** | 6.769 | 0.5547 | 0.7906 | **8.900** | 7.050 |
| Ours | **0.9244** | **0.9316** | 7.725 | **8.350** | **0.9555** | **0.9467** | 7.100 | **9.025** |

- Y-Gradient Cosine Similarity (**Y-GCS**), where we convert both images to YUV, keep only the luminance channel $Y$, compute Sobel gradient magnitudes, normalize them by the shared global maximum, and take the cosine similarity between the two gradient maps, capturing global alignment of edge and fine-detail patterns while being largely insensitive to color changes.

- Y-Gradient Magnitude Similarity (**Y-GMS**), where on the same luminance gradients we compute a per-pixel Gradient Magnitude Similarity (GMS) (Xue et al., 2013) map and average it over the image, providing a more localized and standard gradient-based IQA measure (Wang & Bovik, 2006) of structural consistency.

### B.5.2  EDITING-QUALITY METRICS

Beyond the standard CLIP similarity used in PIE-Bench, we observe that CLIP-based scores often exhibit over-saturated preferences and suffer from inaccurate attribute binding (Kang et al., 2025), making them suboptimal for disentangling the success of color-specific edits. Instead, following recent instruction-based image editing works (Liu et al., 2025b; Wu et al., 2025a; Deng et al., 2025a; Wu et al., 2025b; Cui et al., 2025), we adopt the VIEScore metrics (Ku et al., 2024) from GEdit-Bench (Liu et al., 2025b), which are originally designed to evaluate a wide range of instruction-based editing tasks, including color editing. Therefore, we directly reuse their definitions without any modification. Concretely, we use:

- Semantic Consistency (**SC**), which measures whether the intended edit has been correctly applied.

- Perceptual Quality (**PQ**), which reflects the naturalness of the result and the absence of artifacts or over-saturation.

Each score ranges from 0 to 10 (higher is better) and is produced by a state-of-the-art multimodal LLM evaluator, GPT-4o[2] (Hurst et al., 2024). For video inputs, we uniformly sample three frames per clip, compute **SC** and **PQ** for each frame, and report the mean over the three frame-wise scores as the final scores for that video.

In combination, the structural-preservation metrics (**Y-GCS**, **Y-GMS**) diagnose whether edits preserve structure and details, while **SC** specifically evaluates whether the color edit is successful and **PQ** captures potential over-saturation or unnatural editing artifacts.

---

[2]API access as of Nov. 2025

Table 11: Additional quantitative video results compared with baselines on PIE-Bench.

| Method | Y-GCS ↑ | Y-GMS ↑ | SC ↑ | PQ ↑ |
|---|---|---|---|---|
| Fix seed | 0.5537 | 0.6532 | 4.375 | 6.708 |
| FireFlow (Deng et al., 2025b) | 0.8278 | 0.8419 | 2.508 | 6.967 |
| RF-Solver (Wang et al., 2025) | 0.8517 | 0.8545 | 2.542 | 6.850 |
| SDEdit (Meng et al., 2022) | 0.9167 | 0.9102 | 0.833 | 6.167 |
| DiTCtrl (Cai et al., 2025) | 0.7027 | 0.8396 | 4.933 | 6.258 |
| UniEdit-Flow (Jiao et al., 2025) | 0.8446 | 0.8599 | 3.317 | 6.867 |
| Ours | **0.9383** | **0.9299** | **7.550** | **6.975** |

Table 12: Quantitative real image results compared with training-free models on PIE-Bench.

| Method | Canny | BG Preservation | | Clip Similarity ↑ | | Y-GCS ↑ | Y-GMS ↑ | SC ↑ | PQ ↑ |
| | SSIM ↑ | PSNR ↑ | SSIM ↑ | Whole | Edited | | | | |
|---|---|---|---|---|---|---|---|---|---|
| PnpInversion (Ju et al., 2024) | 0.6178 | 21.60 | 0.7785 | **26.10** | **21.41** | 0.8142 | 0.8106 | 3.650 | 6.575 |
| FlowEdit (Kulikov et al., 2024) | 0.6524 | 21.46 | 0.8324 | 25.71 | 21.05 | 0.7463 | 0.8222 | 3.275 | 7.700 |
| Ours | **0.7683** | **23.87** | **0.8890** | 25.88 | 20.52 | **0.9366** | **0.8969** | 5.450 | 8.425 |

### B.5.3 RESULTS

Tabs. 9 and 11 report additional quantitative metrics on PIE-Bench for SD3, FLUX.1-dev, and CogVideoX-2B, comparing our method against training-free baselines. Across all settings, our approach consistently achieves the highest **Y-GCS** and **Y-GMS** scores, further highlighting its superior ability to preserve edge structure and local details under strong color edits. In contrast, the low **SC** scores of the other baselines indicate that they rarely perform the intended color changes correctly, which is consistent with the qualitative comparisons in the main paper. This also shows that CLIP similarity alone is insufficient to reveal our advantage specifically on color editing.

Tab. 10 compares our method with commercial models. Although they can obtain relatively high **SC**, their lower **PQ** scores support our observation that they tend to over-edit and introduce over-saturated, unrealistic edits. Their low Y-GCS and Y-GMS further demonstrate that they fail to maintain structural and textural consistency.

Overall, these results show that existing training-free methods do not handle color editing well, highlighting that accurate color editing is a challenging problem. Our method is able to achieve high-quality color changes while strongly preserving structure without training.

### B.6 MORE RESULTS ON REAL IMAGE EDITING

We further compare our method with the latest MM-DiT-based FlowEdit (Kulikov et al., 2024) and the classic U-Net–based PnP-Inversion (Ju et al., 2024) on real images from PIE-Bench. For both baselines, we observe that obtaining reasonable results requires carefully tuning which attention layers to edit and which timesteps to intervene at. The optimal choices **vary across backbones** (SD1.5, SDXL, SD3, FLUX.1-dev) and **settings** (noise-to-image vs. real-image editing), making their robustness and usability rather limited. For fairness, we therefore report results using the official implementations with their recommended default hyperparameters.

In contrast, our method can be applied to real images with exactly the same configuration as in the noise-to-image setting: we simply edit **all layers and all timesteps**, without any changes. To the best of our knowledge, this is the **first** attention-control editing method that **does not require any hand-tuned hyperparameters when changing backbones or settings**.

Quantitatively, Tab. 12 shows that our approach consistently and substantially outperforms both FlowEdit and PnP-Inversion on structural-preservation metrics, including **Canny-SSIM**, **Y-GCS**, and **Y-GMS**, as well as background-preservation metrics. While PnP-Inversion sometimes achieves higher CLIP similarity, this must be interpreted with caution in light of the discussion in Sec. B.5.2: CLIP-based scores often exhibit over-saturated preferences and cannot reliably capture artifacts in-

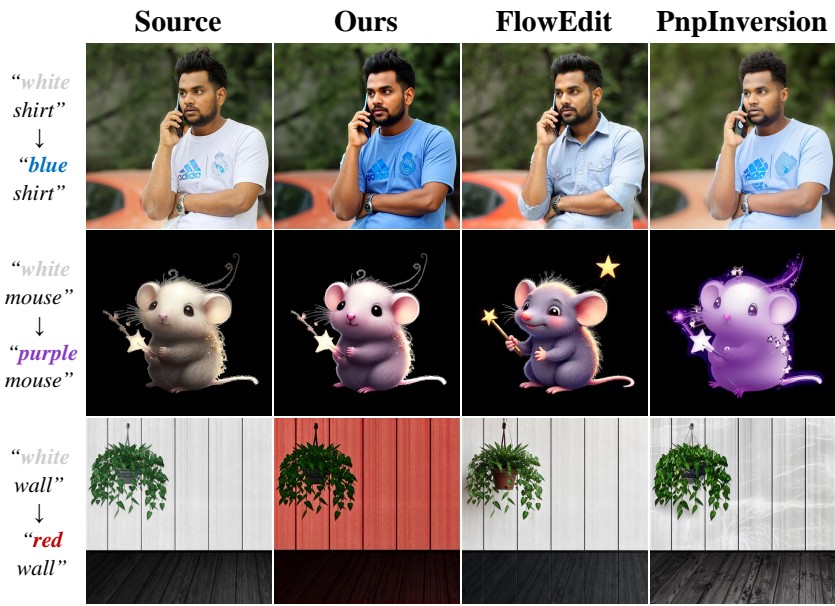

Figure 12: Qualitative real image results compared with training-free models on PIE-Bench.

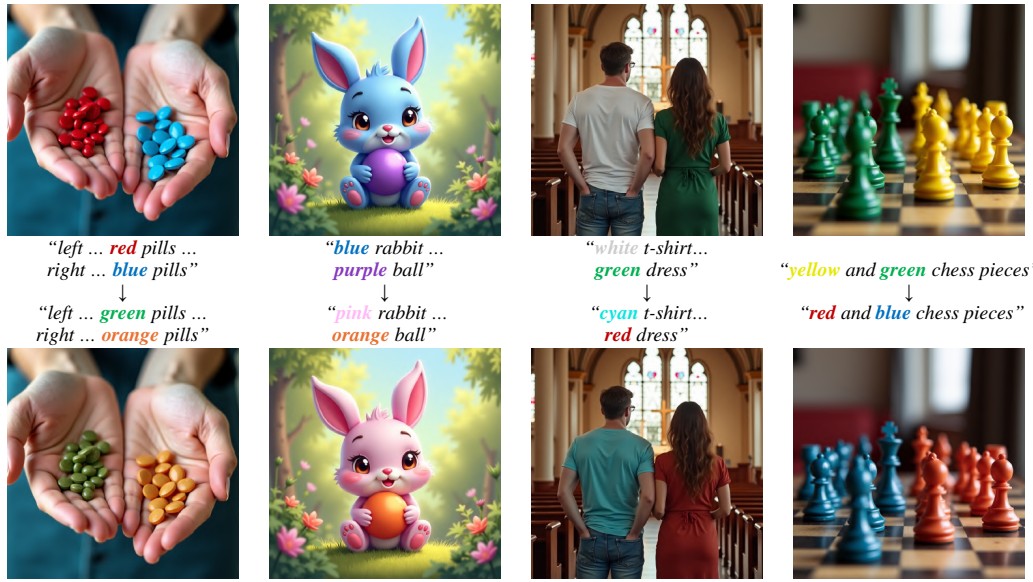

Figure 13: Examples of multi-object editing cases.

troduced by edits. Once we consider **SC** and **PQ**, it becomes clear that PnP-Inversion in fact performs poorly at changing colors as instructed and often produces visually unnatural results, as illustrated in Fig. 12. Overall, our method remains state-of-the-art for real-image color editing, matching the strong structural preservation seen in the noise-to-image setting while achieving significantly better edit quality without any modification of the inference settings.

### B.7 MORE RESULTS OF MULTI-OBJECT EDITING

In addition to the first case in Fig. 1 and the last case in Fig. 11, which already demonstrate our multi-object editing ability, we provide further examples in Fig. 13. As shown, our method can robustly handle scenes with many objects, such as the first example where two separate piles of pills, each

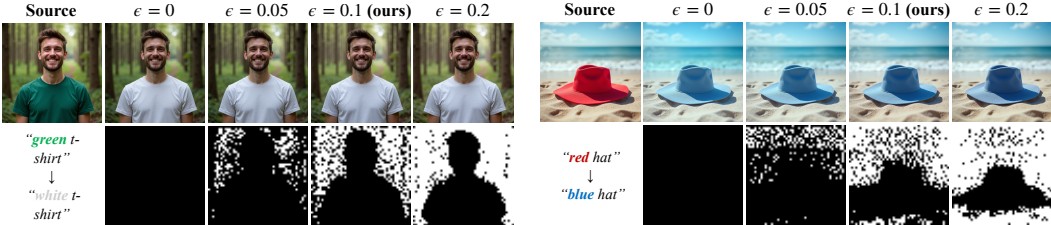

Figure 14: **Ablation study of the mask extraction threshold $\epsilon$.** For better visibility, we visualize $1 - $ mask.

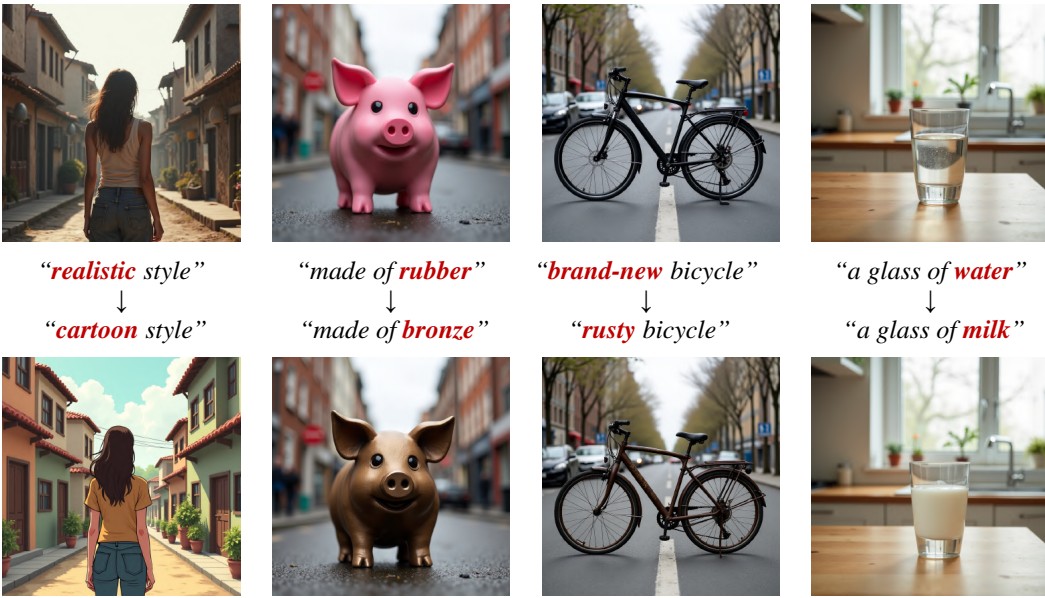

Figure 15: Examples of additional applications enabled by reducing the preservation strength.

containing many pills of a different color, are edited simultaneously. In the second example, multiple objects are spatially close or even overlapping, yet our method still correctly assigns different target colors to each object. In the final example, we not only edit multiple chess pieces but also accurately update their corresponding reflections on the board, further illustrating precise multi-object editing.

### B.8 ABLATION STUDY OF MASK THRESHOLD

The threshold $\epsilon$ for mask extraction is the only hyperparameter in ColorCtrl. We conduct an ablation study in Fig. 14 to evaluate the sensitivity of our method to $\epsilon$. As shown, the performance is **not very sensitive to this parameter**: around our default choice $\epsilon = 0.1$, using a stricter threshold ($\epsilon = 0.2$) or a more relaxed one ($\epsilon = 0.05$) leads to only minor visual differences. Even when we completely remove the mask extraction and color preservation modules ($\epsilon = 0$), the structural details of the background remain unchanged. The main effect of using too small a threshold is a slight hue shift in some background regions, but the overall results are still visually coherent.

### B.9 MORE APPLICATION AND ABILITY

ColorCtrl is designed to edit color while keeping geometry material and light-matter interaction strictly fixed. However, this does not mean it can only produce extremely conservative edits. In fact, by appropriately reducing the number of effective timesteps of structure preservation (*i.e.*, weakening the preservation strength), we can obtain a variety of different applications and effects while still preserving structure, as illustrated in Fig. 15, including style transfer, material change,

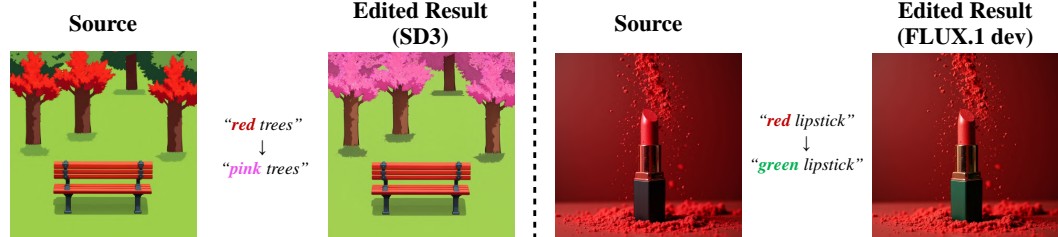

Figure 16: **Examples of failure cases.** Left: results generated with SD3. Right: results generated with FLUX.1-dev.

texture change, and transparency change. In the first example, a real-image style is turned into a cartoon style while preserving geometry extremely well, without requiring a separately trained ControlNet (Zhao et al., 2023). In the second example, a smooth rubber pig is transformed into a metallic bronze pig, showing characteristic metallic texture and reflections while its shape remains unchanged. In the third example, rust-like textures are added to an originally brand-new bicycle. In the last example, we change the transparency of the liquid in a glass, turning clear water into opaque milk, while keeping both the glass structure and the non-edited regions well preserved.

### B.10    LIMITATIONS

The generation quality and the precision of text-guided localization in our method are fundamentally limited by the capabilities of the underlying generative models. As shown in Fig. 16, a failure case with SD3 demonstrates that the model fails to correctly detect "red trees" and instead modifies all trees, including green ones that should remain unchanged. Similarly, in the example with FLUX.1-dev, the model misinterprets "lipstick" and edits the casing rather than the lipstick itself. As foundation models continue to improve, we expect the performance and applicability of our method to advance accordingly.

Furthermore, editing real images and videos remains challenging due to the limitations of current inversion and reconstruction techniques. Although our method performs reliably on data that lies within the training distribution of the generative model, handling real-world inputs requires first mapping them accurately into the latent space of the model. This mapping process is still difficult and highly sensitive to the quality of the inversion.

## C    LARGE LANGUAGE MODEL (LLM) USAGE

We used an LLM to correct grammar, to draft the web UI for the user study, and to generate the ColorCtrl-Bench prompt list. All outputs were manually reviewed and edited by the authors.

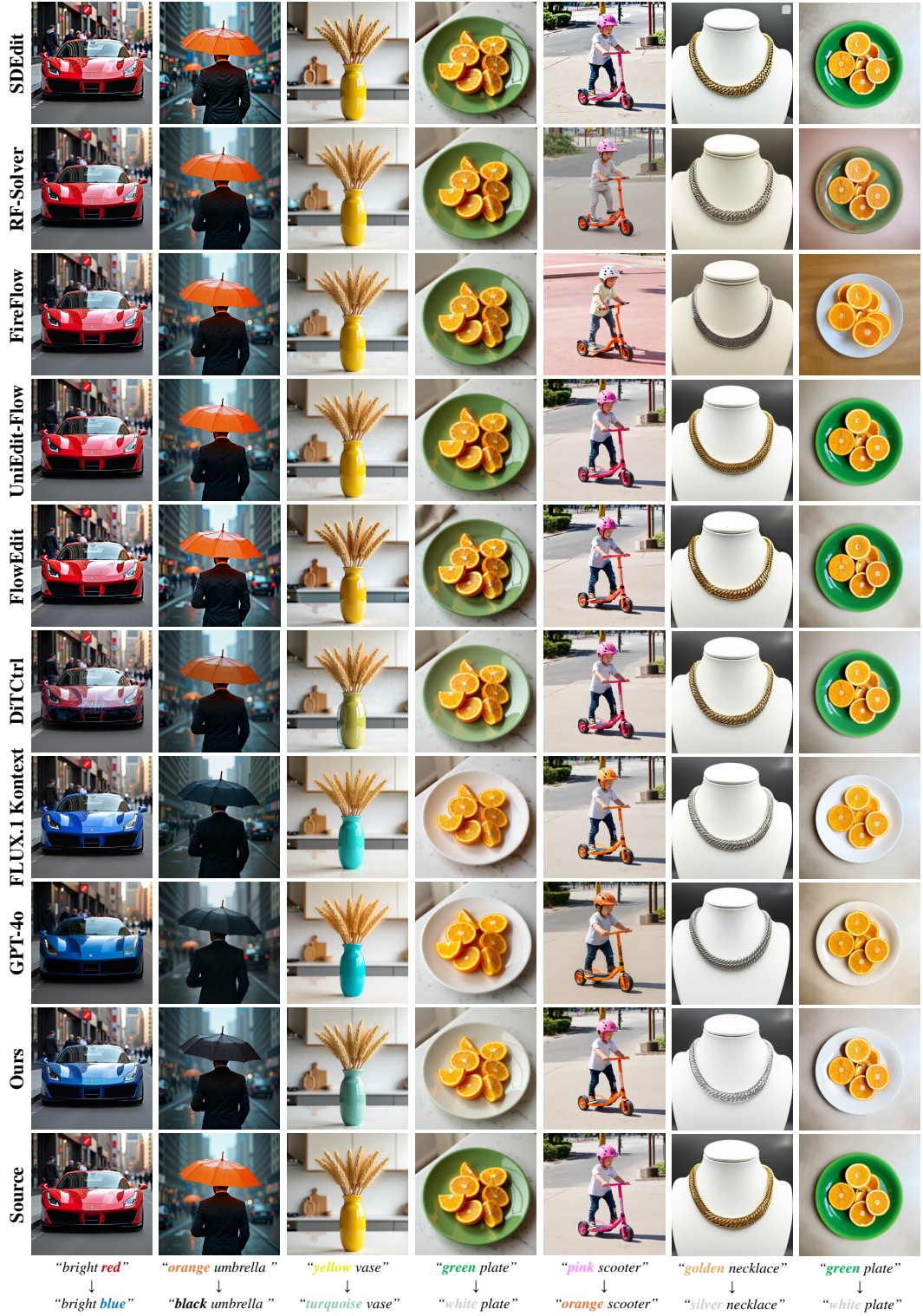

Figure 17: **Qualitative image results compared with training-free methods and commercial models on ColorCtrl-Bench.** The left four columns are generated using FLUX.1-dev, while the right three are generated using SD3. ***Best viewed with zoom-in.***

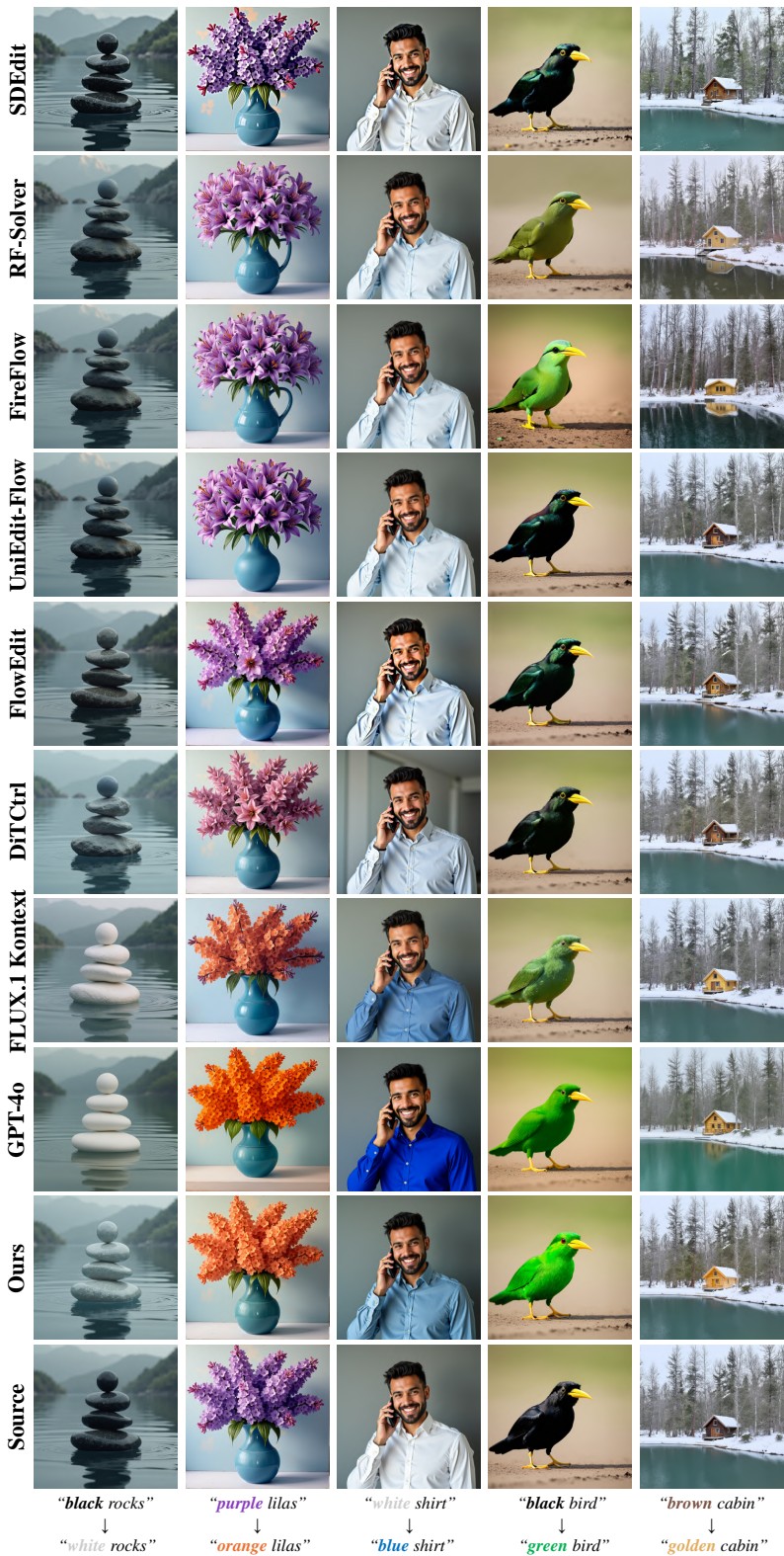

Figure 18: **Additional qualitative image results compared with training-free methods and commercial models on PIE-Bench.** The left three columns are generated using FLUX.1-dev, while the right two are generated using SD3. ***Best viewed with zoom-in.***

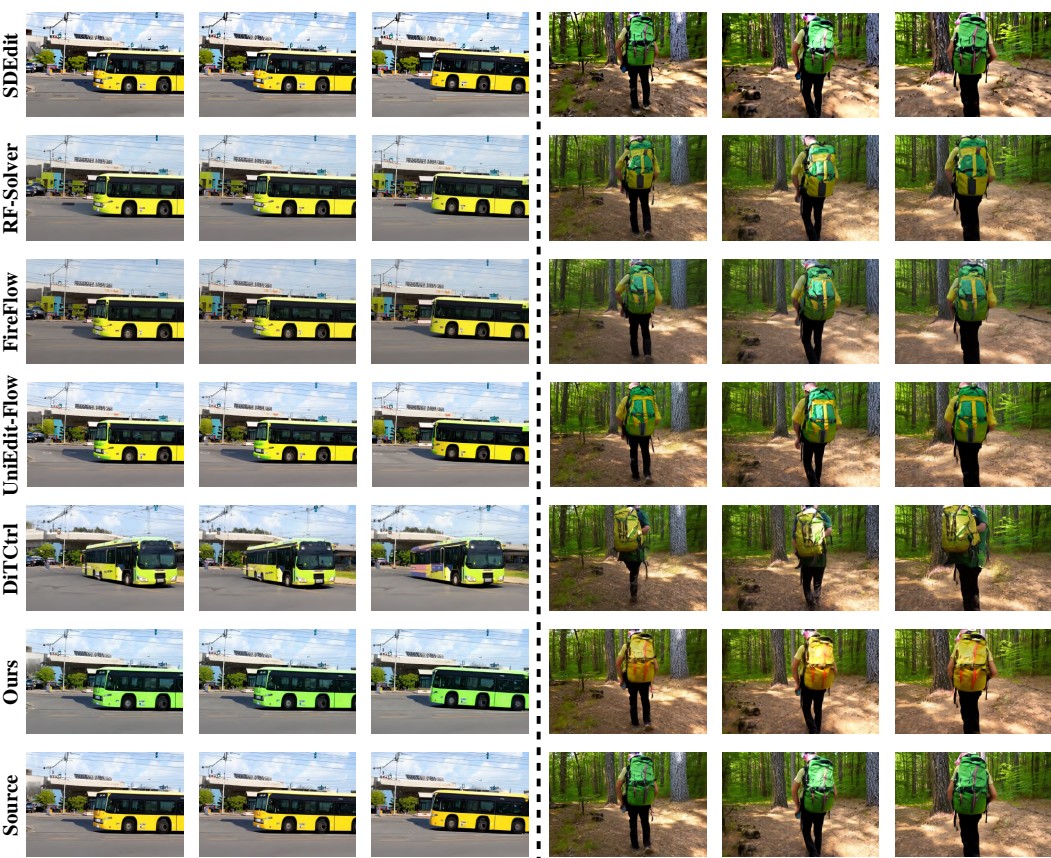

Figure 19: **Qualitative video results compared with training-free models on ColorCtrl-Bench.** Left: "yellow bus" → "green yellow". Right: "green backpack" → "yellow backpack". Each shows three frames.

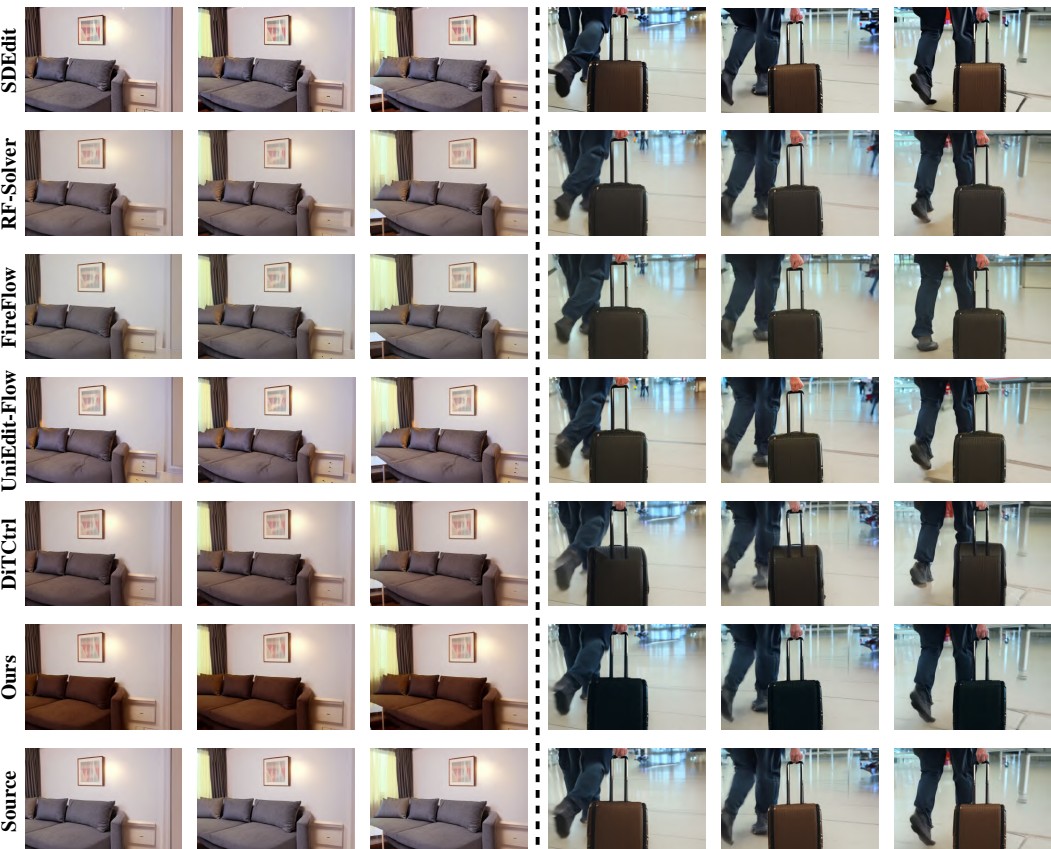

Figure 20: **Additional qualitative video results compared with training-free models on ColorCtrl-Bench.** Left: "gray sofa" → "brown sofa". Right: "brown suitcase" → "black suitcase". Each shows three frames.

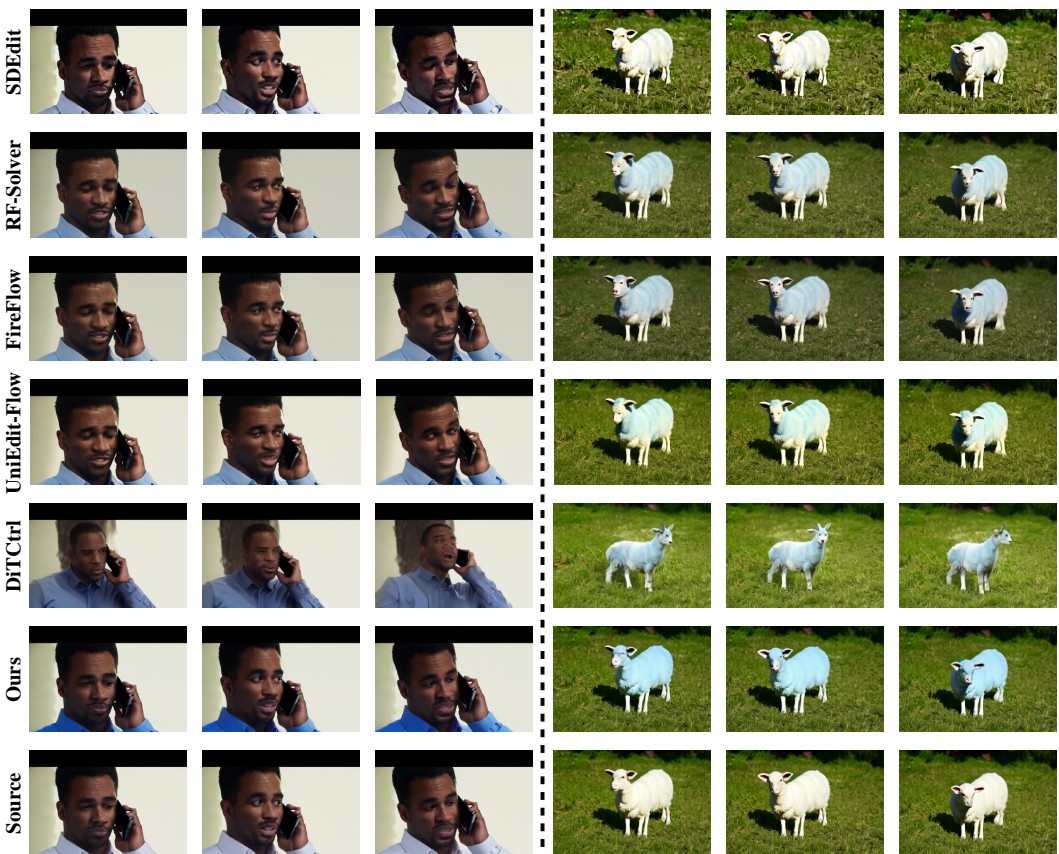

Figure 21: **Additional qualitative video results compared with training-free models on PIE-Bench.** Left: "white shirt" → "blue shirt". Right: "white lamb" → "blue lamb". Each shows three frames.

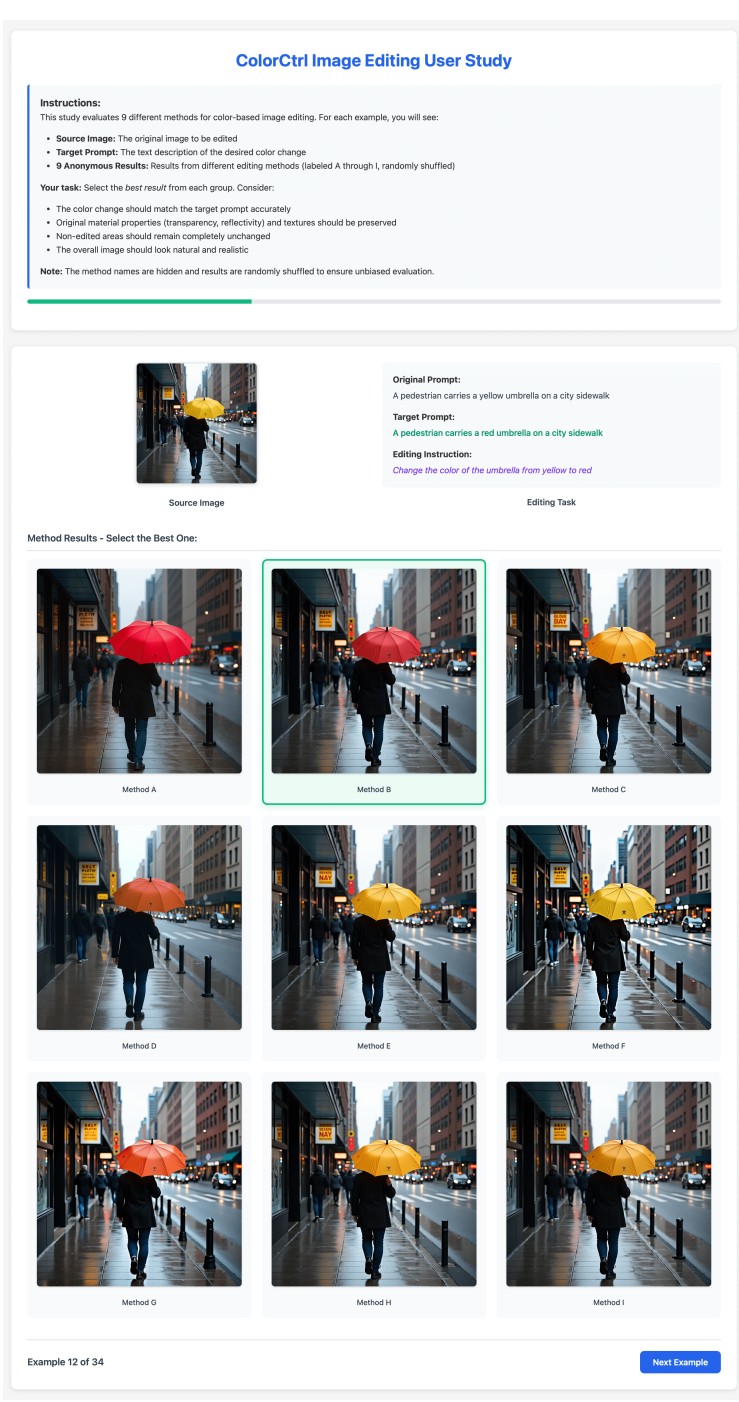

Figure 22: User interface for user study.

Please generate a JSON list of 300 sets. Each set consists of: a source prompt, a target prompt, a instruction, and a subject token.

The source prompt describes a source image.

The target prompt describes the source image after the color of an object has been changed.

The instruction is a description of what needs to be changed to go from the source to the target prompt.

The subject word is the noun that refers to the object that changed color, a single word that appears in the source prompt.

Here is an example:

{

    "src_prompt": "A person wearing a blue shirt is sitting on a chair",

    "tgt_prompt": "A person wearing a yellow shirt is sitting on a chair",

    "subject_word": "shirt",

    "instruction": "Change the color of the shirt from blue to yellow",

}

Only generate examples where there is clearly only one possible object to be changed, so it can be tagged correctly. Write it as a JSON list yourself. Please DO NOT write code; Return only the JSON list.

Figure 23: The prompt used to generate ColorCtrl-Bench with GPT.

