# OpenReview forum: "Training-Free Text-Guided Color Editing with Multi-Modal Diffusion Transformer"
_ICLR.cc/2026/Conference — ICLR 2026 Poster_

### Official Review · Reviewer_45WZ · 2025-10-20

**Soundness:** 3
**Presentation:** 3
**Contribution:** 3
**Rating:** 6
**Confidence:** 4

**Summary:**

The paper presents ColorCtrl, a training-free, text-guided color editing method for images and videos built on Multi-Modal Diffusion Transformers (MM-DiT). The core idea is to disentangle “what should stay” (structure, materials, lighting geometry) from “what should change” (albedo or light color) by (i) swapping the vision-to-vision quadrant of the attention map from a source branch to a target branch to preserve geometry and view, (ii) extracting an edit mask from vision-to-text attention and copying vision value tokens outside the mask to preserve non-edited colors, and (iii) enabling word-level attribute intensity control by pre-softmax scaling in the text-to-vision attention region. Extensive comparisons on SD3 and FLUX.1-dev, plus CogVideoX for video, indicate state-of-the-art training-free performance on PIE-Bench and a new ColorCtrl-Bench; the method reportedly surpasses strong commercial systems (FLUX.1 Kontext Max, GPT-4o Image Generation) on consistency while remaining model-agnostic and compatible with instruction-based editors (Step1X-Edit, FLUX.1 Kontext dev).

**Strengths:**

* Originality: The paper adapts attention-control editing to MM-DiT with a clear decomposition of attention quadrants: vision-to-vision for structure preservation, vision-to-text for mask extraction, and text-to-vision for controllable attribute strength. This differs from U-Net cross-attention methods and prior MM-DiT controls (e.g., DiTCtrl) by operating directly on attention maps and value-token routing without training.

* Quality: The mechanism is well specified: two-branch unrolling with cached source features; mask from vision-to-text attention; copying only vision value tokens for non-edited areas; and pre-softmax token-specific scaling for attribute re-weighting. Ablations show that structure preservation and color preservation cumulatively improve consistency metrics with minimal loss in CLIP alignment.

* Clarity: The paper provides a task formulation grounded in a rendering-style decomposition (G, L, A, S, C), detailed pipeline figures, and explicit implementation/inference settings (steps, CFG, mask threshold, fixed seeds). These details aid reproducibility.

**Weaknesses:**

* Masking and subject detection reliance: The evaluation and parts of the pipeline hinge on subject keywords and a fixed attention-threshold ($\epsilon=0.1$) for mask extraction; robustness to threshold choice, ambiguous subject words, or multi-object scenes is not deeply analyzed.

* Claims versus limitations: The paper acknowledges failures when the base model mislocalizes targets or confuses attributes (e.g., trees or lipstick casing). More systematic characterization of such failure modes—especially under crowded scenes, glossy materials, or colored lighting—would be valuable.

**Questions:**

* How sensitive is performance to the mask threshold ϵ and to the choice of the “blended word” subject token? Please provide a small sensitivity analysis (e.g., ±0.05 around 0.1) and results for ambiguous subjects or multi-instance scenes.

* The benchmark uses noise-to-image generation to isolate editing. Can you complement this with a real-image inversion benchmark that reports reconstruction error and edit success jointly, to reflect common editing workflows?

---

> ### Author Response · Authors · 2025-11-19
> **Response to Reviewer 45WZ (Part 1/N)**
>
> ## Response 1: Mask extraction details and multi-object editing
> Thank you for the suggestion. In addition to Fig. 3a and Sec. 4.6, we have added an ablation study in Appendix B.8 and Fig. 14, which further confirms that coarse masks have negligible impact on output quality. Even without any mask extraction or color-preservation module, ColorCtrl still maintains strong structural fidelity with only mild color shifts. Therefore, a loose 0.1 threshold is sufficient for all settings.
>
> We also provide additional multi-object editing experiments. Beyond the original examples (Fig. 1 left and Fig. 11 right), Appendix B.7 now includes extensive multi-object cases, including scenes with a large number of objects, interacting objects, and objects with mutual reflections. These results further demonstrate that ColorCtrl handles complex multi-object scenarios robustly.
> ## Response 2: Clarification of limitations
> Our failure cases are shown only to illustrate the **typical** failure mode, misidentifying objects or editing slightly beyond the intended region. Importantly, even in difficult scenarios, the model does **not** exhibit other failure types such as texture corruption, background inconsistency, or structural distortion. Crowded scenes do **not** often fail; our paper includes many complex successful examples. For instance, in Fig. 1: the second example handles reflections and transparency correctly; the third handles multi-object interaction under significant day-to-night scene changes; and the fourth shows that even numerous tiny droplets on glass consistently change color with the edited ball. Empirically, we also observe that FLUX performs significantly better than SD3, and we expect the editable range to expand further as base models continue to improve.

---

> ### Author Response · Authors · 2025-11-19
> **Response to Reviewer 45WZ (Part 2/N)**
>
> ## Response 3: Real Image Editing
> Thank you for the thoughtful comments. We first clarify how training-free editing methods are typically categorized and evaluated. Existing works can be roughly grouped into two families.
> 1. **Attention-control methods that reveal new generative capabilities** (e.g., Prompt2Prompt, MasaCtrl, and DiTCtrl). These works generally **do not** report benchmark metrics, because their primary contribution is to demonstrate previously unknown abilities of the underlying pre-trained model in the noise-to-image setting.  Their evaluation focuses on showing the emergence of new forms of controllability, and later works build on these capabilities for downstream tasks besided directly adopted in real image editing, such as drag-based editing (DragDiffusion, FastDrag, Inpaint4Drag), consistency-driven content generation (StoryDiffusion, CharaConsist), or dataset construction for training instruction-based editing models (InstructPix2Pix). ColorCtrl also belongs to this category. It reveals an entirely **new** form of editing capability that has never been exhibited by either U-Net– or MM-DiT-based generative models. Namely, **color edits under extremely strong structural consistency**, preserving geometry, material properties, and even light–matter interactions with hair-level precision. This is a qualitatively new phenomenon rather than an incremental improvement over prior methods.
> 2. **Inversion-based methods that use attention control on real images** (e.g., PnP-Inversion, RF-Solver, and FireFlow). They **do** report benchmarks because their objective is to combine attention control with inversion and validate whether the combined pipeline performs similarly on real images and synthetic images.
>
> From this perspective, **ColorCtrl does not strictly require benchmark results**. Our contribution is the discovery and analysis of a new attention-control capability. Following the convention of prior attention-control papers, it would have been sufficient to show applicability to real images without benchmark results. In the paper, Sec. 4.7 and Fig. 7 already demonstrate real-image and video editing, and Appendix B.1 and Fig. 10 show that ColorCtrl can generate large-scale color edit pairs, data that are extremely challenging to collect in real scenarios, which can be directly used to train instruction-based editing models. This already illustrates significant practical value independent of benchmarks.
>
> Moreover, evaluating the noise-to-image setting is the clearest way to reveal the intrinsic capability of an attention-control method, as it avoids confounding factors from inversion. As shown in Figs. 4, 6, and 17–21, existing training-free baselines fundamentally **fail** to perform strong-consistency color editing. This suggests that ColorCtrl is not simply “better” but rather the **first method that enables this capability at all**.
>
> Nevertheless, we added real image editing comparisons in Appendix B.6. Due to time constraints, we compare with the latest MM-DiT–based FlowEdit (ICCV 2025 best student paper) and the classic U-Net–based PnP-Inversion on real images from PIE-Bench. Importantly, our method applies to real images **with exactly the same configuration** as in noise-to-image editing: we edit all layers and timesteps without changing any hyperparameters. To our knowledge, ColorCtrl is the **first** attention-control editing method that requires **no** hand-tuning when changing backbones (SD3, FLUX.1-dev) or settings (noise-to-image vs. real-image editing). In contrast, baselines require different hyperparameter schedules across backbones and settings, limiting their robustness and usability. Consistent with noise-to-image results, ColorCtrl still outperforms both baselines in qualitative and quantitative comparisons.

---

> > ### Comment · Reviewer_45WZ · 2025-11-24
> > **Response to Authors**
> >
> > Thank you for the detailed response and clarifications. The new analyses and experiments largely address my original concerns, so I will maintain my current score.

---

> > > ### Author Response · Authors · 2025-11-26
> > > **Appreciation for the constructive comments**
> > >
> > > We sincerely appreciate your helpful suggestions! Thanks for your time and effort!

---

### Official Review · Reviewer_QUHF · 2025-10-31

**Soundness:** 3
**Presentation:** 3
**Contribution:** 2
**Rating:** 4
**Confidence:** 4

**Summary:**

This paper introduces ColorCtrl, a training-free framework for text-guided color editing using Multi-Modal Diffusion Transformers (MM-DiT). The method leverages attention maps from MM-DiT to extract semantic masks and reweight attention for localized color edits. The authors claim improved controllability and consistency without requiring model fine-tuning.

**Strengths:**

1. Originality: The idea of using MM-DiT attention maps for color-specific editing is novel in the context of training-free pipelines.The integration of word-level control adds granularity not commonly seen in prior works.
2. Quality: The method is well-implemented and evaluated across multiple domains (images, videos, instructions). Results show high semantic consistency and localized edits, outperforming several baselines.
3. Clarity: The paper is generally well-written, with clear motivation and structured methodology. Visual examples and comparisons are effective in illustrating the method’s capabilities.
4. Significance: The approach is impactful for real-world applications where retraining is costly or infeasible. It contributes to the growing field of training-free multimodal editing, pushing the boundaries of what can be achieved with pre-trained models.

**Weaknesses:**

1. Limited Novelty: While the use of MM-DiT is novel for color editing, similar pipelines have applied attention-based editing in other contexts. TextCrafter also leverages attention maps from MM-DiT to extract semantic masks and reweight attention for image editing. Add-It also leverages attention maps from MM-DiT to extract semantic masks.
2. Insufficient Analysis of Key Components:
(1) The mask extraction process is underexplained: How are attention maps selected? What thresholding strategy is used? How robust is the mask across different prompts?
(2) The attention reweighting mechanism lacks detailed discussion: How is the reweighting computed?
(3) A more thorough ablation study would help isolate the contribution of each component (e.g., mask quality, reweighting strategy).

**Questions:**

1. Could you elaborate on how the semantic mask is extracted from MM-DiT attention maps? Which layers are used? What criteria or thresholds are used to binarize or localize the mask?
2. How is the attention reweighting applied during inference? Is it uniform across all heads/layers or selectively applied?
3. How sensitive is the editing quality to the mask threshold and prompt?

---

> ### Author Response · Authors · 2025-11-19
> **Response to Reviewer QUHF (Part 1/N)**
>
> ## Response 1:  Novelty
> Thank you for acknowledging the novelty of using MM-DiT attention maps for color-specific editing, and we appreciate Reviewer 45WZ’s comments on the originality of our method. We would like to restate our main contribution. ColorCtrl introduces a training-free image and video editing method on MM-DiT that performs text-guided color editing with high-precision structural preservation through vision-to-vision attention maps. This resolves the long-standing trade-off between consistency with the source image and the strength of the edit, which all prior approaches fundamentally fail to overcome. As a result, the structure-preservation mechanism is the core focus of our method.
>
> Beyond this capability, ColorCtrl offers several practical advantages:
> - **No hyperparameter tuning.** The same “all layers, all timesteps’’ configuration works for SD3, FLUX.1-dev, CogVideoX-2B, and real-image editing. Prior methods (TextCrafter, DiTCtrl, Add-It) require backbone- and setting-specific tuning.
> - **Optional fine control.** The reweight slider adjusts only color intensity (e.g., the ambiguity of “dark”) while preserving all structural details. In contrast, reweighting in DiTCtrl and TextCrafter changes the underlying structure of objects (see Fig. 9 in DiTCtrl).
> - **Stable backgrounds.** Our color-preservation module is barely sensitive to mask quality and works with a loose mask weight (0.1). Even without mask extraction or color preservation, ColorCtrl still maintains the structure with only mild color deviations (Fig. 3a, Sec. 4.6, Appendix B.8, Fig. 14).
>
> Regarding related work, several conceptual distinctions are important.
> - **Mask extraction.** DiTCtrl extracts masks using the additional text-to-vision part (L241; Fig. 3b), which reduces mask accuracy. Our method uses the vision-to-text part only, which yields more precise masks, though we only require very coarse masks. Add-It depends heavily on external segmentation models (SAM). More importantly, Add-It modifies Q/K/V (mixing target Q with source K/V), whereas ColorCtrl transfers the entire vision-to-vision attention map from source to target. This is conceptually and operationally different and, to our knowledge, unexplored in prior work.
> - **Attribute reweighting.** TextCrafter (cited in the revision) modifies attention after softmax, similar to DiTCtrl, violating the normalization constraint of attention (L123) and unintentionally altering both edited and non-edited regions (see Fig. 9 in DiTCtrl). ColorCtrl strictly follows the standard attention formulation and adjusts only color intensity while fully preserving structure during reweighting (Fig. 5), a behavior not observed in previous methods.
> ## Response 2: Mask extraction and reweighting details
> We appreciate your suggestion, which allowed us to further elaborate on and emphasize the implementation details. For mask extraction, we use only the **vision-to-text** attention scores, unlike DiTCtrl which additionally relies on the text-to-vision part. Concretely, we locate the text tokens corresponding to the edited object, aggregate their attention scores across **all layers and timesteps**, average them, and threshold the result at **0.1** (the same threshold used in DiTCtrl as shown in L767). We find that this coarse mark works robustly for settings, eliminating the need for layer/timestep selection required by prior methods and substantially improving usability.
>
> Our reweighting differs fundamentally from prior work. We scale only the **text-to-vision** logits of the target text token (e.g., “dark”) before the softmax operation. This location is compatible with operations used in TextCrafter (tanh) while we adopt a simple additive coefficient that guarantees the desired effect. In contrast, both TextCrafter and DiTCtrl modify attention **after** softmax, violating the normalization constraint of attention weights and potentially harming stability.
>
> Finally, Fig. 3a, Sec. 4.6, and the newly added ablation study in Appendix B.8 and Fig. 14 show that coarse masks have negligible impact on output quality. Even without any mask extraction or color-preservation, our method still maintains good structural preservation with only mild color deviations. Thus a loose 0.1 threshold suffices throughout. We emphasize that reweighting is **not** the core contribution of ColorCtrl. It is merely an optional extension built on our precise attention-control framework. Even simple additive scaling already produces good results in Fig. 5 and more advanced variants (e.g., TextCrafter’s tanh) can be explored in future work. Our main strength remains **structure-preserving** editing, with reweighting providing an additional interactive slider while keeping structure intact.

---

> ### Comment · Reviewer_QUHF · 2025-11-24
>
> I still have concerns regarding the novelty.
>
> For mask extraction, attention-based segmentation has been explored in prior works, and similar ideas have been extended from UNet-based diffusion models to DiT-based models. Add-It’s GitHub implementation also demonstrates multiple localization strategies beyond the one described in its paper. In my experience with Add-It, the attention masks are reasonably good. Given this, it is unclear whether the proposed mask extraction mechanism introduces a fundamentally new concept or differs significantly from these existing implementations.
>
> Regarding attribute reweighting, the improvement appears incremental. From the description, the approach seems to offer only minor gains compared to prior methods such as TextCrafter and DiTCtrl.

---

> ### Author Response · Authors · 2025-11-25
> **Response to Reviewer QUHF**
>
> We thank the reviewer for the careful reading. We would like to kindly clarify again that **Mask Extraction and Attribute Reweighting are not listed in our core contributions**. The main novelty of ColorCtrl is the synergy of two mechanisms that together resolve the **long-standing structure–color trade-off**:
>
> 1. **Vision-to-Vision (V2V) for Structure**: Transferring the V2V part of the attention map from source to target preserves the consistency of geometry, material, and light–matter interaction.
> 2. **Value (V) Tokens for Color**: V-token swapping maintains the appearance of non-edited regions with strong color fidelity.
>
> This is fundamentally different from prior work:
> - Add-It blends Keys and Values using hand-crafted ratios.
> - DiTCtrl transfers Keys and Values with hand-designed layer and timestep selections.
> - ColorCtrl, in contrast, transfers V2V and V-tokens across **all layers and all timesteps**, yielding a different and more robust disentanglement.
>
> Because V-token swapping is highly effective, only a coarse mask is required and the mask extraction serves only as **an auxiliary trigger**. **Even without mask extraction**, ColorCtrl preserves structure with only mild color deviations as shown in Fig. 3a, Sec. 4.6, Appendix B.8, and Fig. 14.
>
> Additionally, our reweighting is **NOT** an incremental tweak, even though it is only an add-on for more feasible user interface, which is not listed in our contribution.
> 1. It is a principled mathematical correction rather than a heuristic.
> 2. It functions jointly with V2V and V-token swapping, reinforcing our structure–color disentanglement strategy, which is the core contribution.
> 3. Prior methods such as DiTCtrl and TextCrafter exhibit pronounced structural collapse under reweighting, far beyond small contour deviations, often altering subject identity or hallucinating background content as shown in **Fig. 9 in DiTCtrl**. Our method **avoids these issues** and preserves geometry, material, and lighting (Fig. 5).
>
> To emphasize the difference with DiTCtrl, we cite the paragraph from L121:
> > ColorCtrl differs in key ways: DiTCtrl only applies mask extraction during long video generation, not editing, while our improved method is more robust. Additionally, DiTCtrl’s re-weighting disrupts attention consistency, causing inconsistent geometries, and TextCrafter exhibits a similar issue, while ours avoids this. Furthermore, ColorCtrl works across all layers and timesteps, unlike DiTCtrl, which requires careful layer selection. Different from DiTCtrl, we operate in attention maps rather than key and value tokens.
>
> ## **TL;DR**
> **Novelty goes beyond isolated technical modifications. What matters is whether a method advances the community’s capabilities and practice. ColorCtrl provides such progress. As shown in Fig. 1, our training-free approach achieves results that, to the best of our knowledge, have not been demonstrated by existing academic or commercial methods.**

---

> > ### Author Response · Authors · 2025-11-27
> > **Thank you for the review!**
> >
> > Thank you for taking the time to review our work and for the thoughtful attention you have given to our submission.
> >
> > We hope you have had an opportunity to read our response submitted on November 19 and 25. In these responses, we provided detailed descriptions of our auxiliary modules, including mask extraction and attribute reweighting, along with additional ablation studies on mask localization.
> >
> > We would also like to emphasize that, to the best of our knowledge, our work is the **first** to introduce the Vision-to-Vision component for structure preservation and the value tokens for color preservation within MM-DiTs. These modules enable color-editing capabilities that are difficult to achieve with existing training-free or commercial models, as illustrated in Fig. 1.
> >
> > In addition, attention-control techniques have previously expanded the capabilities of generative models beyond their original design. For example, key and value tokens swapping in MasaCtrl has enabled methods such as story generation (StoryDiffusion, CharaConsist), interactive drag-based editing (DragDiffusion, FastDrag), long video generation (DiTCtrl), and object insertion (Add-It). Our method follows a similar spirit by providing a simple yet generalizable mechanism that broadens the expressive range of MM-DiTs in the domain of targeted color editing.
> >
> > If you have any other questions or concerns, we would be very happy to offer further clarification.
> >
> > We appreciate your thoughtful consideration of our work.
> >
> > The Authors

---

### Official Review · Reviewer_Ld1H · 2025-11-01

**Soundness:** 2
**Presentation:** 3
**Contribution:** 2
**Rating:** 4
**Confidence:** 4

**Summary:**

This paper proposes a training-free color editing method for images and videos based on pretrained T2I/T2V MMDiT models. It disentangles the color and structure within the edited region, and achieves accurate and natural color editing without touching unrelated regions or attributes. The proposed method is evaluated on extensive examples with various base models, surpassing strong commercial models. A new benchmarking protocol is proposed by adopting the PIE-bench prompts on generated images and adjusted metrics. The proposed method also applies to videos and instructional image editing models seamlessly.

**Strengths:**

- The proposed method is dedicatedly designed for color editing and exclude other factors, achieving clean color editing results with faithfully maintained content.

- The proposed method and benchmark highlights the importance of producing reasonable color edit with similar lighting etc. environmental conditions, over the traditional standard semantic CLIP alignment, making the output results more realistic and natural.

**Weaknesses:**

- [Major] The only editing quality metric is still CLIP similarity, which doesn't align with the claim that CLIP usually leads to over saturated preference as it lacks details. The major upgrade of metrics only focus on the structure preservation. Given the focus of the paper, some improved editing metrics are necessary to solidate the evaluation. For example, at least aesthetics/harmony can be tested to show the edited colors fit in the environment well. Maybe other color spaces like HSV could also help to decompose? Texture preservation can also be considered (e.g. semantic not impacted), while Canny is relatively rough especially when the threshold is not low enough. For CLIP similarity, it could also potentially help if it is calculated separately between color words and object words etc.

- [Major] The proposed method is tested mainly on generated images instead of real images, and the proposed benchmark also emphasizes this. Although a dedicated section is provided for application on real images, the major quantitative results are compared on synthetic data. It is claimed that testing on generated images can calibrate the impact of inversion quality etc., but in practice it's still important to evaluate on real data, in terms of both performance and methodology. The performance of all methods will be impacted by inversion, while some methods might cooperate better or worse with inversion, and the difference matters for real usage.

- UNet-based models sometimes feature their tighter spatial correspondence over transformers. One or two strong baselines could be involved.

**Questions:**

- For the failure cases described in Sec. B.5, would there be additional approach to improve? For example, would it be feasible to locate the editing region with the full object in the source image caption, i.e. use "red trees" or "red lipstick" instead of "trees" or "lipstick" to locate the desired region?

- Would the proposed method be applied for other attribute editing like color that are uniform and doesn't change structures, e.g. textures, reflection/transparency? If so the scope of this paper could be largely extended.

---

> ### Author Response · Authors · 2025-11-19
> **Response to Reviewer Ld1H (Part 1/N)**
>
> ## Response 1: Evaluation Metrics
> Thank you for the helpful comments on evaluation metrics. In the revision, we add both **structural-preservation** and **editing-quality** metrics, and evaluate all baselines (training-free and commercial) on PIE-Bench using SD3, FLUX.1-dev, and CogVideoX-2B.
> - **Structural-preservation metrics.** In the original submission, we followed the common practice in the ControlNet literature and used **Canny-SSIM** to assess structural consistency. Based on your suggestion, we now introduce two finer-grained metrics: **Y-Gradient Cosine Similarity (Y-GCS)** and **Y-Gradient Magnitude Similarity (Y-GMS)**, which are both computed on the Y channel in YUV space. We do not compute the V channel in HSV since it is still strongly affected by color and contrast. Hence, Y channel is more suitable than V channel in our setting. Details are provided in Appendix B.5.1.
> - **Editing-quality metrics.** Previously, we used the standard metrics defined by the original benchmarks without modification. For better aesthetic/quality measures, we additionally adopt the VIEScore metrics from GEdit-Bench, which have recently been widely used in image editing works such as Qwen-Image, EMU3.5, OmniGen2, Bagel, and Step1X-Edit. Since the evaluation of color-editing has been designed in VIEScore metrics, we use its definitions without any changes. In particular, **SC (Semantic Consistency)** measures whether the requested edit is correctly followed, and **PQ (Perceptual Quality)** measures whether the resulting image is natural and coherent, capturing issues such as over-saturation and editing artifacts. For video, we uniformly sample three frames per clip, compute SC and PQ on each frame, and report the arithmetic mean as the video-level score. Details are in Appendix B.5.2.
>
> Appendix B.5.3 reports results for all baselines across the three backbones (image and video). The **Y-GCS** and **Y-GMS** scores highlight the strong ability of ColorCtrl to preserve structural consistency. In contrast, other training-free baselines have low **SC**, showing that they almost never correctly perform the intended color edits, which is consistent with qualitative results. Commercial models often achieve high **SC**, but their low **PQ** supports our observation that they tend to over-edit and introduce noticeable distortions. Also, the additional structural-preservation metrics also indicate that they fail to maintain fine details. These results confirm that structure-consistency color editing is a challenging task, and that ColorCtrl achieves high-quality color changes without training.

---

> ### Author Response · Authors · 2025-11-19
> **Response to Reviewer Ld1H (Part 2/N)**
>
> ## Response 2: Real Image Editing
> Thank you for the thoughtful comments. We first clarify how training-free editing methods are typically categorized and evaluated. Existing works can be roughly grouped into two families.
> 1. **Attention-control methods that reveal new generative capabilities** (e.g., Prompt2Prompt, MasaCtrl, and DiTCtrl). These works generally do not report benchmark metrics, while their primary contribution is to demonstrate previously unknown abilities of the underlying pre-trained model in the noise-to-image setting.  Their evaluation focuses on showing the emergence of new forms of controllability, and later works build on these capabilities for downstream tasks besided directly adopted in real image editing, such as drag-based editing (DragDiffusion, FastDrag, Inpaint4Drag), consistency-driven content generation (StoryDiffusion, CharaConsist), or dataset construction for training instruction-based editing models (InstructPix2Pix). ColorCtrl also belongs to this category. It reveals an entirely **new** form of editing capability that has never been exhibited by either U-Net– or MM-DiT-based generative models. Namely, **color edits under extremely strong structural consistency**, preserving geometry, material properties, and even light–matter interactions with hair-level precision. This is a qualitatively new phenomenon rather than an incremental improvement over prior methods.
> 2. **Inversion-based methods that use attention control on real images** (e.g., PnP-Inversion, RF-Solver, and FireFlow). They **do** report benchmarks because their objective is to combine attention control with inversion and validate whether the combined pipeline performs similarly on real images and synthetic images.
>
> From this perspective, our contribution is the discovery and analysis of a new attention-control capability, rather than the comparisons on benchmark. Following the convention of prior attention-control papers, it would have been sufficient to show applicability to real images without benchmark results. In the main body of this paper, Sec. 4.7 and Fig. 7 already demonstrate real-image and video editing, and Appendix B.1 and Fig. 10 show that ColorCtrl can generate large-scale color edit pairs, data that are extremely challenging to collect in real scenarios, which can be directly used to train instruction-based editing models. This already illustrates significant practical value independent of benchmarks.
>
> Moreover, evaluating the noise-to-image setting is the clearest way to reveal the intrinsic capability of an attention-control method, as it avoids confounding factors from inversion. As shown in Figs. 4, 6, and 17–21, existing training-free baselines fundamentally **fail** to perform strong-consistency color editing. This suggests that ColorCtrl is not simply “better” but rather the **first method that enables this capability at all**.
>
> Nevertheless, we added real image editing comparisons in Appendix B.6. Due to time constraints, we compare with the latest MM-DiT–based FlowEdit (ICCV 2025 best student paper) and the classic U-Net–based PnP-Inversion on real images from PIE-Bench. Importantly, our method applies to real images **with exactly the same configuration** as in noise-to-image editing: we edit all layers and timesteps without changing any hyperparameters. To our knowledge, ColorCtrl is the **first** attention-control editing method that requires **no** hand-tuning when changing backbones (SD3, FLUX.1-dev) or settings (noise-to-image vs. real-image editing). In contrast, baselines require different hyperparameter schedules across backbones and settings, limiting their robustness and usability. Consistent with noise-to-image results, ColorCtrl still outperforms both baselines in qualitative and quantitative comparisons.
> ## Response 3: Comparison with U-Net-based Baselines
> Thank you for the thoughtful suggestion. A fair noise-to-image comparison across backbones is not possible, because different backbones produce different source images and therefore cannot be evaluated on identical inputs. Prior works (RF-Solver, FireFlow) already show that MM-DiT is strictly stronger than U-Net in this setting, so adding U-Net noise-to-image baselines would not be meaningful.
>
> In the **real-image** setting, all methods share the same input, and we compare with U-Net baseline PnP-Inversion in Appendix B.6. ColorCtrl outperforms it with more consistent structure preservation and more natural editing results.

---

> > ### Author Response · Authors · 2025-11-27
> > **Thank you for the review!**
> >
> > Thank you for taking the time to review our work and for the thoughtful attention you have given to our submission.
> >
> > We hope you have had an opportunity to read our response submitted on November 19. In that response, we provided four additional evaluation metrics across three models, a new benchmark evaluation on real image editing with MM-DiT-based and additional U-Net–based baselines, as well as further ablation studies on mask localization and additional attribute editing, including style, material, texture, and transparency changes.
> >
> > If you have any other questions or concerns, we would be very happy to offer further clarification.
> >
> > We appreciate your thoughtful consideration of our work.
> >
> > The Authors

---

> ### Author Response · Authors · 2025-11-19
> **Response to Reviewer Ld1H (Part 3/N)**
>
> ## Response 4: Mask Localization and Limiations
> Thank you for the suggestion. The issues you observed are mainly caused by the capability of the base model. Our failure cases are shown only to illustrate the **typical** failure mode, misidentifying objects or editing slightly beyond the intended region. Importantly, even in difficult scenarios, the model does **not** exhibit other failure types such as texture corruption, background inconsistency, or structural distortion. All examples are directly taken from PIE-Bench without any hand-crafted prompt optimization, and better prompts can indeed help the model find the mask. Empirically, we also observe that FLUX performs significantly better than SD3, and we expect the editable range to expand further as base models continue to improve. In real applications, external mask extractors such as SAM can also be used. Even without any mask extraction or color-preservation, our method still maintains good structural preservation with only mild color deviations (Fig. 3a, Sec. 4.6, Appendix B.8, Fig. 14).
>
> Our paper includes many complex successful examples. For instance, in Fig. 1: the second example handles reflections and transparency correctly; the third handles multi-object interaction under significant day-to-night scene changes; and the fourth shows that even numerous tiny droplets on glass consistently change color with the edited ball.
> ## Response 5: Extending ColorCtrl Beyond Color Editing
> We appreciate your comment as it encouraged us to showcase a broader range of editing scenarios. Our goal is to highlight the strong fine-detail preservation enabled by ColorCtrl. This level of consistency can be relaxed to support other editing behaviors that require similar structural fidelity, such as style change, material change, texture change, and transparency change. In practice, this is achieved by applying the control mechanism only in the early timesteps. We provide examples in the revised version (Appendix B.9, Fig. 15).

---

### Author Response · Authors · 2025-11-19
**General response to reviewers**

We thank all reviewers for their valuable and insightful feedback. We are encouraged that the reviewers found our work to be **novel and original** (Reviewer QUHF, 45WZ),  recognized that our **state-of-the-art** results show high consistency and localized edits, outperforming several baselines and even strong commercial models (Reviewer Ld1H, QUHF, 45WZ), and acknowledged the **impact** on pushing the boundaries of real-world applications (Reviewer QUHF) as well as the **detailed task formulation** grounded in a rendering-style decomposition (Reviewer 45WZ).

Our work has benefited substantially from your feedback. Below we summarize the main modifications to the manuscript (highlighted in blue in the revised PDF):
1. **Additional evaluation metrics.** In response to Reviewer Ld1H’s comments, we added two structure-focused metrics, Y-Gradient Cosine Similarity (Y-GCS) and Y-Gradient Magnitude Similarity (Y-GMS), to more finely measure detail and structural preservation under color edits. We also incorporated Semantic Consistency (SC) and Perceptual Quality (PQ) from GEdit-Bench to better evaluate the success of the edits and the naturalness of the results. We evaluated all baselines, including both training-free and commercial models, on image and video editing tasks on PIE-Bench with SD3, FLUX.1-dev, and CogVideoX-2B. These results further demonstrate that our method produces more effective, more natural, and more structurally consistent edits than that of competing approaches (Appendix B.5).
2. **Additional benchmark (real-image editing).** Building on the suggestions of Reviewer Ld1H and 45WZ, in addition to the existing noise-to-image setting, we added a real-image editing benchmark on PIE-Bench (Appendix B.6). We compare against the latest MM-DiT-based method FlowEdit (ICCV 2025 best student paper) and the classic U-Net–based method PnP-Inversion. Besides the standard evaluation metrics, we also report the newly added metrics above. The results show that our method remains highly effective in the real-image editing setting as well.
3. **Additional qualitative results.** Following the suggestions of Reviewer Ld1H and 45WZ, we added more examples of multi-region, multi-object editing in Appendix B.7. In addition, we showcase other editing capabilities that require strong structural preservation beyond color changes, including style transfer, material changes, texture changes, and transparency changes (Appendix B.9).
4. **Ablation study.** Incorporating feedback from Reviewer Ld1H, QUHF, and 45WZ, we conducted an ablation study on the mask threshold $\epsilon$ in Appendix B.8. The results show that our method is not sensitive to this hyperparameter: even disable mask extraction and color preservation ($\epsilon = 0$), the structural preservation remains highly strong.

We hope these revisions and clarifications address the main concerns raised in the reviews, and we are happy to further discuss any remaining questions. If you find our response satisfactory, we kindly ask you to consider raising your scores in recognition of our core contributions: a training-free, MM-DiT–based attention-control method that substantially advances the challenging task of color editing (for both images and videos) while strongly preserving geometry and structural details across backbones and settings.

---

### Author Response · Authors · 2025-11-30
**Summary of Rebuttal and Discussion**

**Dear Area Chair,**

We would like to provide a summary of our rebuttal process to assist in your final recommendation. All reviewers agree that our work is **novel/original** (QUHF, 45WZ), achieves **state-of-the-art consistency and localized edits**, outperforming several baselines and even strong commercial models (Ld1H, QUHF, 45WZ), and has **clear practical impact** (QUHF) and a **well-structured rendering-style formulation** (45WZ).

During the discussion, we extensively updated the manuscript to address all concerns. **Notably, Reviewer 45WZ has explicitly responded: *"The new analyses and experiments largely address my original concerns."***

Here is a summary of how we resolved key issues and significantly elevated the paper's contribution:

**1. Significantly Extended Scope Beyond Color (Addressing Reviewer Ld1H)**
Reviewer Ld1H insightfully noted that if our method could apply to attributes beyond color (e.g., transparency, textures), **"the scope of this paper could be largely extended."**
*   **Achievement:** We successfully demonstrated this capability. Leveraging our strong structural preservation, we showcased effective editing for **Style Transfer, Material Changes, Texture Editing, and Transparency Adjustments**.
*   **Impact:** This fulfills the reviewer's criteria for a "largely extended scope," transforming our method from a color-specific method into a **general-purpose, structure-aware editing method**.

**2. Validated on Real-Image Editing with "Zero-Tuning" Robustness (Addressing Ld1H, 45WZ)**
Reviewers requested validation on real images to ensure practical utility.
*   **Achievement:** We benchmarked against the latest MM-DiT-based SOTA (**FlowEdit**, best student paper in ICCV 2025) and classic U-Net-based baseline (**Pnp-Inversion**) on PIE-Bench, showing superior performance.
*   **Impact:** Crucially, we demonstrated that ColorCtrl is the **first** attention-control method requiring **NO hyperparameter tuning** across different backbones (SD3, FLUX.1-dev, CogVideoX-2B), modalities (Image/Video), or settings (noise-to-image vs. real-image editing). This "zero-tuning" capability proves superior robustness compared to baselines that require distinct schedules for each setup.

**3. Verified Robustness via Ablation Studies (Addressing Ld1H, QUHF, 45WZ)**
Reviewers inquired about the sensitivity of the mask threshold ($\epsilon$).
*   **Achievement:** We added a detailed ablation study.
*   **Impact:** The results prove high robustness: even with the mask extraction disabled ($\epsilon=0$), our core structure-preservation mechanism effectively maintains geometry. This confirms that our method does not rely on precise masks, addressing concerns about sensitivity.

**4. Superiority Over Baselines via Advanced Metrics (Addressing Ld1H)**
To address concerns about evaluation metrics:
*   **Achievement:** We integrated **structure-focused metrics** (Y-GCS/Y-GMS) and **aesthetic/quality metrics** (VIEScore). We re-evaluated all baselines using SD3, FLUX.1-dev, and CogVideoX-2B, including commercial models on image and video tasks.
*   **Impact:** The results quantitatively confirm that ColorCtrl generates more natural and structurally consistent edits than training-free and commercial models.

**5. Clarified Technical Novelty and Implementation (Addressing QUHF)**
Reviewer QUHF asked for details on auxiliary modules.
*   **Achievement:** We provided in-depth clarifications on mask extraction and reweighting.
*   **Impact:** We emphasized that our core contribution is the **vision-to-vision part swapping for structure preservation** and **v-token swapping for color preservation**, which is fundamentally distinct from prior works. The auxiliary modules are optional enhancements.

**Conclusion**
By extending the paper's scope to general attribute editing, proving robust "zero-tuning" generalization, and strictly validating our advantages over baselines, we believe we have fully addressed the reviewers' concerns. We hope these contributions warrant a positive decision.

Best regards,

The Authors

---

### Meta-Review · Area_Chair_kfTx · 2025-12-28

**Summary:**

This paper proposes a color-editing method for pre-trained image and video models by introducing a training-free approach. The reviewers’ concerns mainly focus on: (1) the novelty of the work, (2) implementation details, and (3) additional evaluations. The authors provided detailed responses clarifying the implementation details (such as mask extraction and multi-object editing), added results across various evaluation metrics, and included more results on real-image editing, which adequately address the corresponding concerns. The remaining concern relates to the perceived limited novelty raised by one reviewer. The authors provided additional explanations to justify the differences between this work and existing methods. Overall, the authors have provided detailed responses to each concern and question, and one reviewer has confirmed that the rebuttal resolved their concerns. The discussion around novelty might not be the reason to reject this paper, and the authors are encouraged to provided clear explanation in the final version about this concern. Given the contributions of this work, the paper is recommended for acceptance.

**Reviewer Concerns:**

I think most of the concerns from reviewers (e.g., additional evaluation metrics, the performance of the method on real-image editing, more implementation details like mask extraction, and discussion of limitations) have been addressed in the rebuttal. The concern regarding novelty, however, remains unresolved between the authors and the reviewers.

**Reviewer Scores:**

Reviewer 45WZ participated in the discussion with the authors and confirmed that the concerns were resolved. The reviewer holds a positive rating.

The other two reviewers hold negative ratings. Reviewer Ld1H did not participate in the discussion with the authors and raised two major concerns. One concern is about how the method performs on real-image editing, which was also raised by Reviewer 45WZ and has been confirmed as resolved by Reviewer 45WZ. The other concern relates to the evaluation metrics. The authors provided additional evaluation metrics requested by the reviewer in the revised appendix. These additional experiments may be sufficient to address Reviewer Ld1H’s concerns. Thus, Reviewer Ld1H may change their score to positive.

Reviewer QUHF participated in the discussion but still expressed concerns regarding the novelty of the work. The authors provided additional information to clarify the novelty of the paper. It remains unclear whether Reviewer QUHF will be convinced.

---

### Decision · Program_Chairs · 2026-01-26

Accept (Poster)